# Sustainable DNA-polysaccharide hydrogels as recyclable bioplastics

Yujie Ke[1,2,11], Kai Lan[3,11], Jing Yi Wong[1,4], Hongfang Lu[1,5], Shujun Gao[6], Keunhyuk Ryu[7], Feng Chen[1,5], Wei Wei Loh[1], Zhili Dong[7], Jason Y. C. Lim[1,8], Zhaogang Dong[1,9], Xi Chen[2] ✉, Itamar Willner[10] ✉ & Yuwei Hu[1] ✉

Traditional petrochemical-derived plastics are challenging to recycle and degrade, and the existing (re)process methods are organic solvent-based and/or energy-intensive, resulting in significant environmental contamination and greenhouse gas emissions. This study presents a sustainable bioplastic material characterized by multi-closed-loop recyclability and water (re)processability. The bioplastics are derived from abundant polysaccharide sources of dextran, alginic acid, carboxymethyl cellulose, and DNA of plant and living organism waste. The process involves chemical oxidation of polysaccharides to produce aldehyde-functionalized derivatives, which subsequently form reversible imine covalent bonds with amine groups in DNA. This reaction yields water-processable polysaccharide/DNA crosslinked hydrogels, serving as raw materials for producing sustainable bioplastics. The bioplastic products exhibit (bio)degradability and recyclability, enabling aqueous recovery of the hydrogel constituents through plastic hydrolysis and the natural biodegradability of DNA and polysaccharides. These products demonstrate excellent resistance to organic solvents, self-healing, scalability, and effective processing down to nanometer scales, underscoring their potential for broad and versatile applications. The work provides potential pathways for advancing sustainable and environmentally friendly bioplastic materials.

The global production of plastics has reached unprecedented levels with annual figures continuing to escalate, with demand expected to double before 2050[1,2]. The predominant source of commercialized plastics is derived from petrochemical resources, a non-renewable commodity[3]. This reliance on finite resources not only perpetuates environmental degradation but also underscores the urgent need for sustainable alternatives[3]. The pervasive use of conventional plastics has elicited profound environmental consequences, encompassing marine pollution, and adverse impacts on wildlife, and contributed significantly to carbon emissions[4,5]. Also, the accumulation of plastic

[1]Institute of Materials Research and Engineering (IMRE), Agency for Science, Technology and Research (A*STAR), 2 Fusionopolis Way, Innovis #08-03, Singapore 138634, Republic of Singapore. [2]School of Interdisciplinary Studies, Lingnan University, Tuen Mun, Hong Kong SAR, China. [3]Department of Forest Biomaterials, North Carolina State University, 2820 Faucette Drive, Raleigh, North Carolina 27695, USA. [4]Department of Biomedical Engineering, National University of Singapore, 4 Engineering Drive 3, Singapore 117543, Republic of Singapore. [5]Institute of Molecular and Cell Biology, Agency for Science Technology and Research (A*STAR), 61 Biopolis Drive, The Proteos, Singapore 138673, Republic of Singapore. [6]National Neuroscience Institute, 11 Jln Tan Tock Seng, Singapore 308433, Republic of Singapore. [7]School of Materials Science and Engineering, Nanyang Technological University, Singapore 639798, Republic of Singapore. [8]Department of Materials Science and Engineering, National University of Singapore, 9 Engineering Drive 1, Singapore 117576, Republic of Singapore. [9]Science, Mathematics, and Technology (SMT), Singapore University of Technology and Design (SUTD), 8 Somapah Road, Singapore 487372, Republic of Singapore. [10]Institute of Chemistry, The Center for Nanoscience and Nanotechnology, The Hebrew University of Jerusalem, Jerusalem 91904, Israel. [11]These authors contributed equally: Yujie Ke, Kai Lan. ✉e-mail: chenxi@ln.edu.hk; itamar.willner@mail.huji.ac.il; ywhu@imre.a-star.edu.sg

wastes and the generated microplastics during the degradation process poses severe threats to the environment and sustainable ecosystems. Among the myriad challenges posed by plastic pollution, two paramount concerns have emerged as focal points of extensive research: the recycling and upcycling of plastics[4,6–11] and the proliferation of microplastics in natural ecosystems and organisms[12–14]. Addressing these multifaceted issues demands a concerted effort to develop innovative solutions that prioritize recyclability, utilize renewable resources, employ low-energy manufacturing processes, as well as promote biodegradability and biocompatibility.

In contrast to plastics based on non-renewable fossil fuels, bioplastics are intrinsically biodegradable, recyclable, and usually derived from renewable biomass sources[3]. There is an emerging interest in both industry and academia to shift the raw material resources from non-renewable fossil fuels to renewable biomass to address environmental issues[15–18], e.g., to reduce greenhouse gas emissions[14,16]. So far, bioplastics face several challenges such as facile methods for the breakdown of polymer chains and non-ecofriendly synthesis processes that involve toxic solvents and/or energy-intensive processes. As the genetic information carrier, DNA is biocompatible, designable, and biodegradable, with interesting applications both in vivo and in vitro[19–22]. Most previous works on these applications require careful DNA sequence design and oligo synthesizer to build specific sequences from nucleotides. According to previous reports[23,24], the global biomass DNA reservoir amounts to roughly 50 billion metric tonnes, and harnessing a small fraction of this (under 1%) could fully supply the annual raw materials demand for worldwide plastic manufacturing. Exploration of biomass DNA-based bioplastics is still in the infancy stage. Recently, two studies reported poly(ethylene glycol) diacrylate (PEGDA)[24] and ionomer (composed of poly(epichlorohydrin- co-ethylene oxide) and 1-butylimidazole)[25] as the 3D networks crosslinking units to prepare DNA hydrogel and bioplastics via chemical or physical interactions, respectively. While highly innovative, the synthesis process could be more sustainable, as it involves the extensive use of organic solvents and relatively harsh synthesis conditions and/or relies on key components derived from unsustainable petroleum feedstocks. These increase their potential carbon footprint and environmental impact.

Here, we develop a universal water-based approach to prepare bioplastics from biomass DNA and polysaccharides. Polysaccharides are renewable, naturally abundant, and biocompatible organic biopolymers naturally engineered by living organisms. The polysaccharides, such as dextran (Dex), carboxymethyl cellulose (CMC), and alginic acid (AA), undergo oxidation to introduce aldehyde groups, which subsequently react with amino groups within the biomass DNA, resulting in the creation of robust three-dimensional crosslinked polymer networks. The resulting bioplastics derived from biomass DNA and polysaccharides possess a range of highly desirable properties, including water-(re)processability, (bio)degradability, multi-closed-loop recyclability, scalability, nanoscale fabrication precision, biocompatibility, edibility, and resistance to organic solvent. Importantly, the multi-closed-loop recycling, and water-(re)processability significantly enhance the sustainability of these bioplastics. Life cycle assessment (LCA) demonstrates that producing 1 m³ of these bioplastics generates 861 kg $CO_2$-eq $m^{-3}$ with recycling 10 times, 58.7%-78.3% lower than commercial plastics without recycling. Additionally, these bioplastics minimize the risks associated with microplastic pollution and reduce carbon emissions throughout their lifecycle. This innovative development holds great promise in addressing environmental challenges, providing a sustainable alternative to conventional petroleum-based plastics.

## Results and Discussion

The bioplastics are produced from naturally abundant biomass DNA (Fig. 1a i) and polysaccharides (Fig. 1a ii), and the polysaccharide Dex is used as an example for Fig. 1a ii. The eco-friendly multi-closed-loop life cycle of the produced bioplastics is illustrated in Fig. 1b, including production, recycling, and end-of-life treatment. These biomass DNA and polysaccharides are from bio-renewable resources that are widely present in creatures, plants, germs, and others (Fig. 1a i)[26]. The chemical reaction and the preparation method are illustrated in Fig. 1b ii and 1b iii, respectively. Taking Dex as an example, the synthetic process involved Malaprade oxidation of vicinal diols using $NaIO_4$ (Fig. 1a ii) and a subsequent Schiff base reaction of the resulting aldehydes with the abundant amine groups of the DNA nucleobases (Fig. 1a i) to form imine bonds under aqueous conditions. Both steps were conducted at room temperature without the requirement of inert gas protection. The hydrogel is characterized by DNA and polysaccharide chains connected through imine bonds (Fig. 1b ii). The imine bonds are applied due to their well-known reversible bonding and de-bonding characteristics in water-based mild conditions[27–29], and such dynamical bindings are widely utilized to endow plastics, including polyester-[30], epoxy-[31], and vanillin-based plastics[32], with chemical recycling capabilities. Previous work to crosslink DNA and ionomers involves the mass use of N,N-dimethylformamide (a 10-to-1 mass ratio to ionomers), a relatively high reaction temperature of 130 °C, and an inert gas environment of nitrogen[24]. In comparison, the reactions in the present study are water-based and proceed at room temperature under ambient atmosphere, thereby reducing the energy consumption and potential negative environmental impacts during the synthesis process. More importantly, the bioplastics exhibit three closed-loops (Fig. 1 b): (1) physical recycling through water-based healing and remolding (Fig. 1b iii), (2) chemical recycling enabled by reversible imine bonds (Fig. 1b ii), and (3) a life-cycle loop spanning bio-renewable resources, product formation, and decomposition into biodegradable waste (Fig. 1b i-iv). This multi-closed-loop feature was not reported in previous DNA-based bioplastics[24,25], while it is important to maximize the recyclability and minimize the environmental threat posed by microplastics. The combination of sustainable manufacturing and excellent recyclability plays a crucial role in reducing negative environmental impacts and lowering carbon emissions, as demonstrated in the following LCA. In addition, the bioplastics are demonstrated with a range of highly desirable attributes in the following discussion, including (bio)degradability, scalability, precision in nanoscale fabrication, biocompatibility, edibility, and resistance to organic solvents (Fig. 1c).

After the oxidation of Dex, highly reactive aldehyde groups are generated, which can conjugate with amines of DNA under mild conditions, resulting in the formation of a Dex-DNA composite. The successful bonding between Dex and DNA chains is supported by nuclear magnetic resonance spectroscopy (NMR) (Fig. S1-S10), Fourier transform infrared (FTIR) spectroscopy (Fig. S11), and rheology measurements (Fig. 2a). In the ¹H NMR spectrum of oxidized Dex (Fig. S1), the aldehyde group shows a characteristic signal at -δ 9.7 ppm[33]. No characteristic signal is observed at this region from natural-derived DNA (Fig. S2). Upon the formation of Dex-DNA, an additional signal is obtained at δ 9.2–9.3 ppm (Fig. S3), suggesting the formation of imine groups (RN = CHR′), which is also observed in recycled sample (Fig. S4). As amine group is on N6, N4 and N2 positions of adenine (A), cytosine (C) and guanine (G), respectively, but not thymine (T) (Fig. 1a i), to determine which bases interact with the aldehyde groups of Dex, DNA oligomers of nine bases (A9, C9 and G9) were used, respectively, to react with Dex. Compared to the ¹H NMR spectra of A9, C9, G9 (Fig. S5, S6, S7) and Dex (Fig. S1), the formed Dex-A9, Dex-C9 and Dex-G9 exhibit additional signals at -δ 9.25, 9.26, 9.27 ppm (Fig. S8, S9, S10), respectively, corresponding to the formation of imine groups.

IR measurements further validate the reversible imine bond formation. In the FTIR spectra (Fig. S11), the −C=O stretching vibration (1650 cm⁻¹, 1631 cm⁻¹) and −N-H stretching vibration (3332 cm⁻¹, 3210 cm⁻¹) are observed on the DNA. Compared to Dex, the oxidized

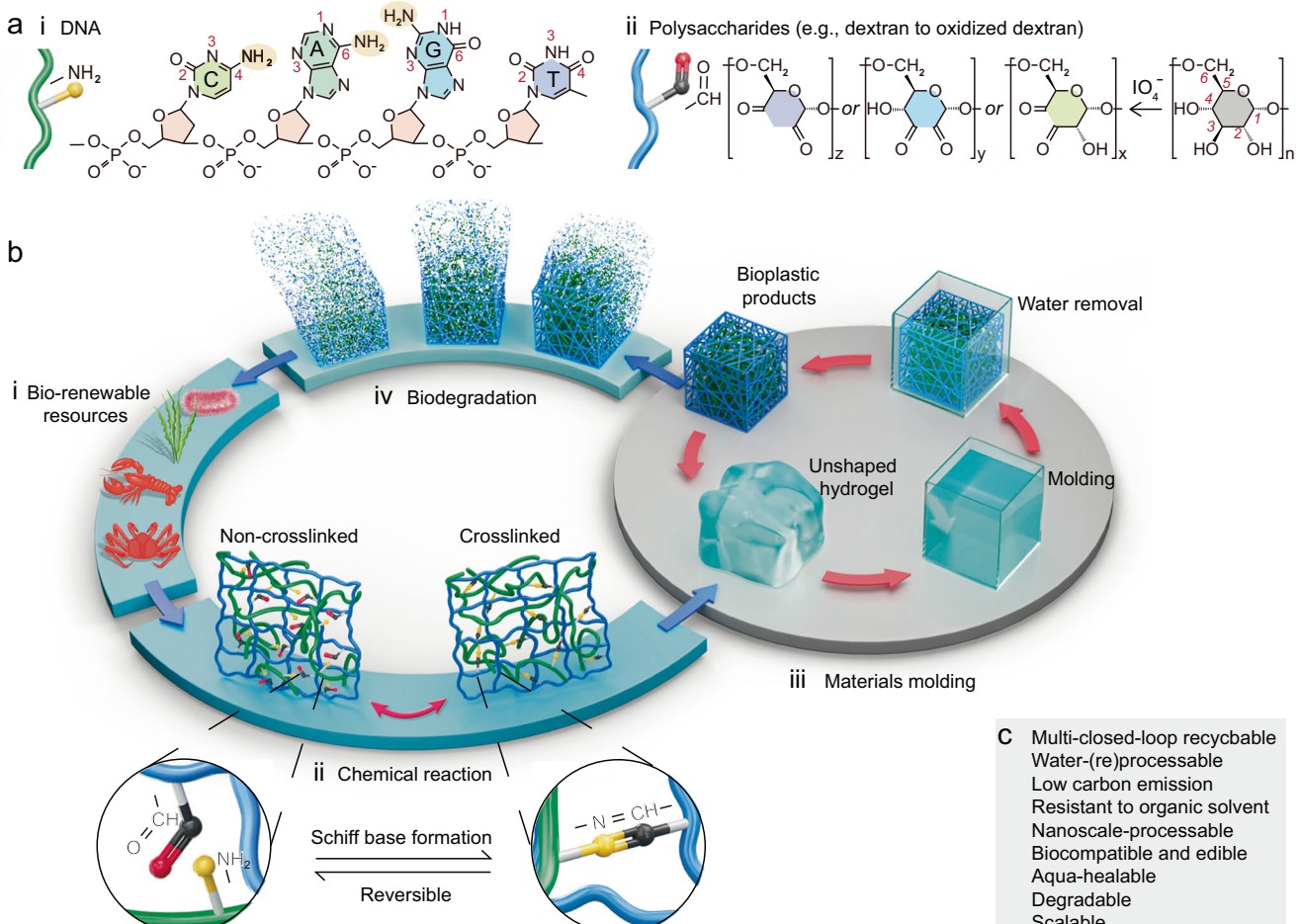

**Fig. 1 | Schematic of the bioplastic and its closed-loop cycles. a** Chemical compositions of biomass (i) DNA and (ii) polysaccharides using dextran and oxidized dextran as examples. **b** Illustration of the multiple closed loop cycles of the DNA-polysaccharides-based bioplastics including the first cycle from (i) bio-renewable resources, (ii) chemical reaction, (iii) materials molding, to (iv) biodegradation, the second cycle of (ii) imine bond-based reversible chemical reaction, and (iii) the third cycle of a water-recyclable molding. **c** Highlights of the bioplastic merits.

Dex exhibits a low-intensity signal at 1732 cm$^{-1}$, corresponding to the −C=O stretching from aldehyde groups[33]. Upon the formation of Dex-DNA composite, −C=N stretching vibration of Schiff base exhibits a characteristic peak at 1654 cm$^{-1}$, which is also observed in the recycled sample. The reversible nature of imine bonds offers significant potential for chemical recycling[27–29], complemented by water-(re)processability under mild conditions, while it compromises the bioplastic stability in humid/acidic environments, as discussed in the following sections. Moreover, we note that reversible imine bonds support chemical recycling, while other factors including chain composition, polymer crystallinity, particle size and environmental factors, such as enzymes, exposure to sunlight and mechanical abrasion (e.g., through wave action), also play key roles in polymer degradation and recycling.

Bioplastic samples are named according to their compositions and the solid weight percentages in the hydrogel state, unless otherwise specified. For example, 10%Dex-5%DNA refers to bioplastics prepared from hydrogels containing 10 wt.% oxidized Dex and 5 wt.% DNA. Rheology study reveals that the 10% Dex behaves as a liquid with G′ ≈ G″ ≈ 0.1 Pa (G′, storage modulus, G″, loss modulus), and the 5% DNA forms a hydrogel with G′ value of ~50 Pa and G″ value of ~20 Pa (Fig. 2a i and ii). In contrast, the formed 10%Dex-5%DNA composite exhibits as a hydrogel with significantly enhanced mechanical properties, showing G′ and G″ values of ~170 Pa and 40 Pa, respectively (Fig. 2a iii). The increased G′ and G″ values indicate a higher degree of crosslinking, attributed to the formation of imine bonds, cooperatively stabilized by H-bonds, between Dex and DNA. The formulation is proven effective for Dex-5%DNA hydrogels across a broad range of Dex concentrations, from 2% to 20% (Fig. 2b i). DNA is shown to be uniformly distributed within the Dex-DNA hydrogel, as evidenced by green fluorescence imaging (Fig. S12). Well-shaped bioplastic products could be derived from hydrogel composites (Fig. 2b ii). Increasing the Dex content in the composite results in hydrogels with higher G′ and G″ values (Fig. S13a). These hydrogels exhibit a reduction in G′ value with elevated temperature from 20 to 80 °C (Fig. S13b). Moreover, the generated Dex-DNA bioplastics can be successfully recycled and re-shaped into various configurations (Fig. 2b iii and iv). Notably, the solid content in the hydrogel plays a crucial role in the shrinkage of generated bioplastics. For example, a bioplastic sample with a 4% solid content (2% Dex and 2% DNA) showed significant shrinkage within 60 minutes after transferring from freeze dryer to atmospheric conditions (Fig. S14), which is ascribed to the lack of sufficient solid to support the porous structures under ambient atmosphere.

A typical closed-loop recycling process is illustrated in Fig. 2c, using a knife-like 10%Dex-5%DNA bioplastic as an example (Fig. 2c i). The bioplastic waste (Fig. 2c ii) is recycled by adding water (Fig. 2c iii), followed by a mixing process to fully dissolve the bioplastics to form a hydrogel (Fig. 2c iv). The recycled hydrogel was used to produce solid bioplastic products through the same freeze-drying and molding process. Both the original and recycled bioplastics are observed with the characteristic imine signals at δ 9.2–9.3 ppm and 1654 cm$^{-1}$ of the

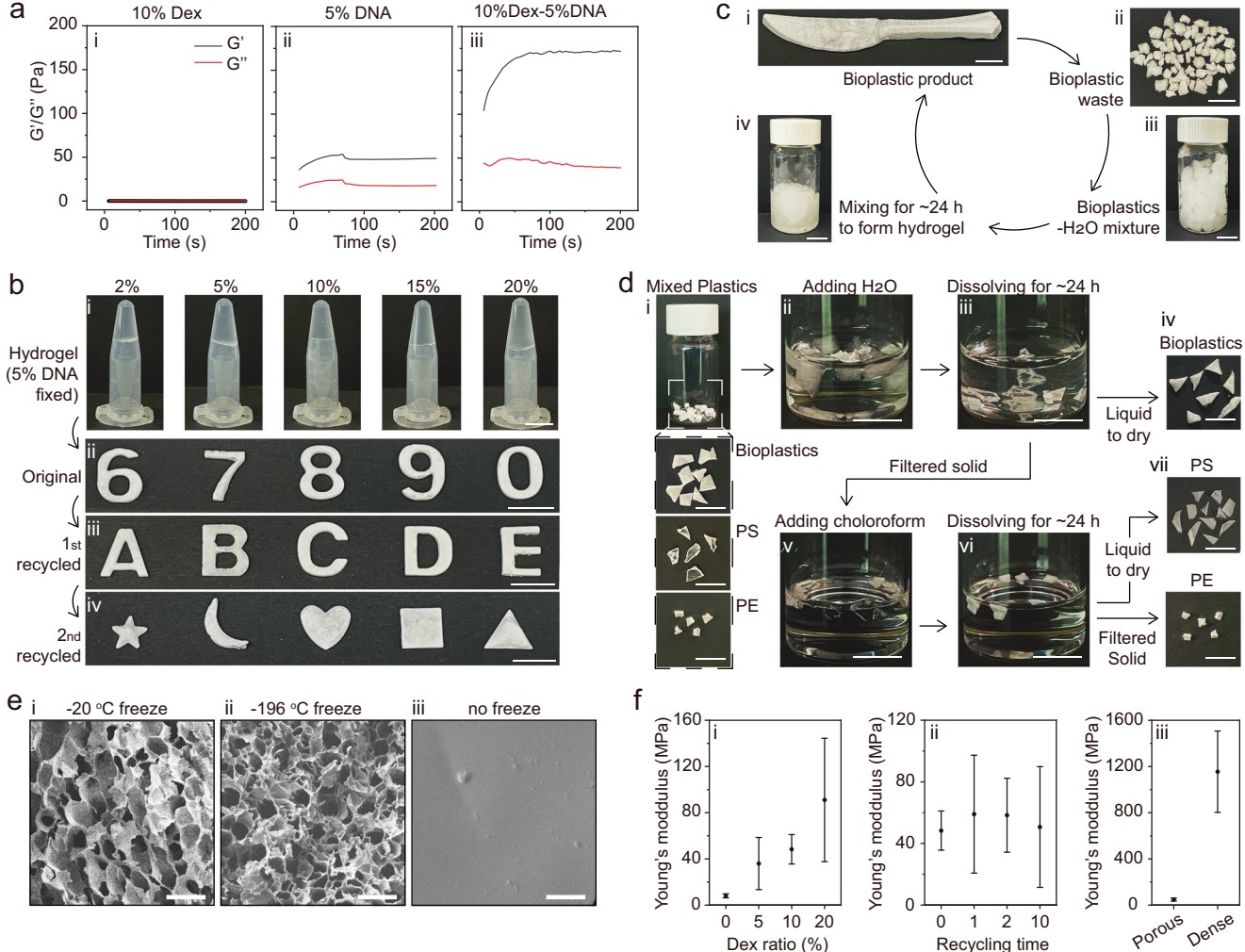

**Fig. 2 | Preparation and recycling of Dex-DNA bioplastics. a** Rheology characterizations of Dex (i), DNA (ii), and Dex-DNA (iii). **b** Photographs of the Dex-5% DNA hydrogels with different Dex contents (i), as well as their bioplastic products for original (ii), 1st (iii), and 2nd (iv) recycling. **c** Illustration of a typical water-processable recycling process. **d** Selective recycling processes for mixed plastics consisting of bioplastics, polyethylene (PE), and polystyrene (PS). **e** Representative SEM images of the bioplastics produced through −20 ℃ (i) and −196 ℃ (ii) freeze-drying, as well as without freeze-drying (room temperature, iii). Three independent experiments were conducted with similar results. **f** Analysis of Young's modulus of the bioplastics with the impact of Dex contents (i), recycling cycles (ii), and microstructures (iii). Data are presented as mean±standard deviation (n = 3). The scale bar is 1 cm in (**b-d**) and 50 µm in (**e**).

−C=N stretching vibration in the $^1$H NMR (Fig. S4) and FTIR spectra (Fig. S11), respectively, suggesting the recyclability. Importantly, the water-processable recycling capability allows the bioplastics to be recovered from mixed plastic wastes, a common challenge in daily plastic recycling. We demonstrate this by mixing the bioplastics with two widely used commercial plastics: polystyrene (PS) and polyethylene (PE) debris (Fig. 2d i). The recycling is carried out through selective solvent dissolution. Water and chloroform are used to selectively recover the bioplastics and PS from the mixed plastics. The bioplastics swell in water (Fig. 2d ii) and fully dissolve after approximately 24 hours (Fig. 2d iii). The solid and liquid components are separated by filtration, and the recycled bioplastic solid is obtained by drying the liquid phase (Fig. 2d iv). The remaining solid is further separated by adding chloroform (Fig. 2d v). The PS plastic debris dissolves completely after ~24 hours (Fig. 2d vi). Recovered PS and PE solids are then obtained from the liquid and solid phases, respectively (Fig. 2d vii).

The freezing and drying process is critical for controlling the microstructures of the produced bioplastics. Scanning electron microscope (SEM) results indicate that the freezing-drying method generates porous microstructures, with bioplastics frozen at −20 ℃

exhibiting larger pores (Fig. 2e i) compared to these rapidly frozen using liquid nitrogen (−196 ℃) (Fig. 2e ii). The bioplastic pore sizes are approximately 10-40 µm (Fig. 2e i) and 2-10 µm (Fig. 2e ii) prepared under −20 ℃ and −196 ℃ conditions, respectively. Moreover, bioplastics dried directly at room temperature display a dense surface with no observable pores at the microscale (Fig. 2e iii). During room-temperature drying, the gradual evaporation of water enables polymer chains within the hydrogel to reorganize and deposit along the mold boundaries, see schematic illustration in Fig. S15. This confined drying promotes the formation of a dense, shaped bioplastic structure, as the capillary forces and polymer interactions guide the hydrogel shrinkage and consolidation within the mold cavity. The dense bioplastics exhibit a stable water vapor transmission rate (WVTR) of ~480-500 g/m²·day and permeability (*P*) of ~200 g·mm/m²·day·kPa on both initial and recycled samples (Fig. S16). The melting point is measured to be 166 ℃ and 160 ℃ in the initial and recycled bioplastics, respectively (Fig. S17), without significant variation.

We performed the tensile tests to investigate changes in the bioplastic's properties throughout the preparation and recycling processes. The results indicate that, with a fixed DNA content of 5%, increasing Dex content from 5% to 20% significantly enhances Young's

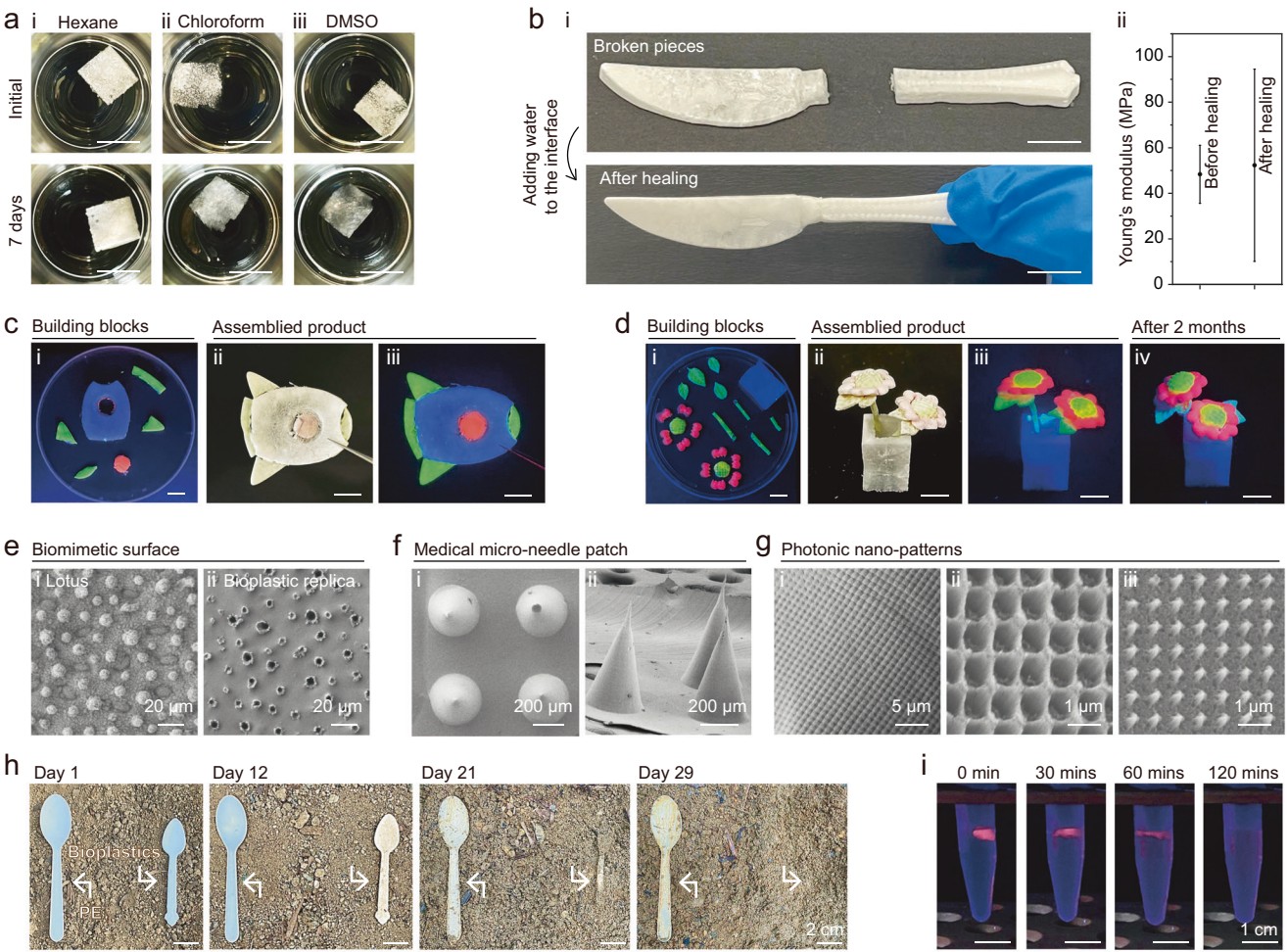

**Fig. 3 | Properties of the bioplastics: chemical resistance, aqua-healing, precise processability, and biodegradation. a** Photographs of bioplastic samples soaked in various chemicals, including hexane (i), chloroform (ii), and dimethyl sulfoxide (DMSO) (iii), at day 0 and day 7. **b** Photographs of an aqua-healing process on a knife-shaped product (i), and Young's modulus analysis before and after aqua-healing (ii). Data are presented as mean±standard deviation (n = 3). Photographs of the assembly of building blocks into a 2D rocket (**c**) and a 3D flower (**d**) under UV irradiation or exposure to natural light. The bioplastics were stained with fluorescent dyes. Representative SEM images of the (**e**) biomimetic surface, (**f**) microneedle patch, and (**g**) photonic nano-patterns. Three independent experiments were conducted with similar results. (**h, i**) Photographs of the biodegradation process of the bioplastics: (**h**) buried in soil under real-world conditions, and (**i**) degraded in the lab using DNase I with GelRed staining to visualize the degradation more clearly. The scale bar is 1 cm in (**a**–**d**).

modulus from ~36 MPa to ~91 MPa (Fig. 2f i and Fig. S18). All Dex-DNA bioplastics exhibit a higher Young's modulus than the pure DNA sample, which has a modulus of ~8 MPa. This demonstrates that Dex-DNA bioplastics possess significantly improved stiffness compared to DNA alone. Additionally, even after 10 times recycling, Young's modulus shows no significant decline (Fig. 2f ii and Fig. S18), highlighting the high recyclability of these bioplastics. Moreover, bioplastics prepared by room-temperature drying (without freezing) exhibit notably higher Young's modulus compared with porous bioplastics (Fig. 2f iii and Fig. S18). For example, using the same 10%Dex-5%DNA formulation, room-temperature dried bioplastics exhibit Young's modulus of ~1155 MPa, approximately 24 times higher than that achieved through −20 °C freeze-drying, ~48 MPa, (Fig. 2f iii). This difference is primarily attributed to the microstructure variation between the dense and porous samples (Fig. 2e). Further comparison between previously reported studies (DNA-based[24,25] and water-processable plastics[5,34−36]) and several typical commercial plastics (Fig. S19)[37] and the presently introduced bioplastics was conducted. It is found that the Young's modulus of the bioplastic is higher than that of DNA-based plastics/gels[24,25], and closely overlaps the values demonstrated by biofilm-/protein-based bioplastics[5,36], high-/low-density polyethylene (HDPE/LDPE)[37], and polyvinylidene fluoride (PVDF)[37]. These results confirm

the bioplastics' high recyclability and demonstrate that their mechanical properties can be fine-tuned by adjusting the freezing process during the preparation process.

We further demonstrate that bioplastics possess several key advantages in terms of chemical resistance and aqua-healing properties. Notably, the bioplastics exhibit intrinsic resistance to several commonly used solvents: they show negligible damages after soaking for 7 days in hexane (Fig. 3a i), chloroform (Fig. 3a ii), and dimethyl sulfoxide (DMSO) (Fig. 3a iii). It is worth mentioning that chloroform can dissolve petroleum-derived PS and low-density polyethylene (LDPE)[11]. The bioplastics exhibit slow dissolution in ethylene glycol or softening/swelling in glycerol and formamide (Fig. S20 and Movie S1-2). While rapid dissolution of bioplastics under extremely acidic (pH ~1, Fig. S21a) and basic (pH ~14, Fig. S21b) conditions is obtained. Additionally, the bioplastics demonstrate excellent aqua-healing capability, illustrated by a knife-shaped bioplastic. After breaking, the fractured pieces are reassembled to their original shape by applying water to the broken interface, followed by a drying process (Fig. 3b i). A subsequent tensile test demonstrates that there is no significant change in Young's modulus before and after the healing process, confirming the high reliability of the aqua-healing feature (Fig. 3b ii). It is worth mentioning that water processability, including water-based molding and aqua-

healing, is a relatively eco-friendly approach. Nevertheless, this indicates the relatively poor moisture resistance of the produced bioplastics. Thus, for practical applications such as packaging materials where high moisture resistance is necessary, further processing of the Dex-DNA bioplastics is required. Potential methods include applying moisture-resistant layers, such as hydrophobic coating (e.g., poly-dimethylsiloxane), to these bioplastics.

The bioplastics are highly processable to form well-shaped products, ranging from centimeter to nanometer scale. The centimeter-scale samples were prepared using the −20 °C freeze-drying method. Beyond the single-piece molding, we also present an effective strategy that assembles building blocks to generate complex 2D and 3D products (Fig. 3c, d). The building blocks are bioplastic pieces stained with DNA intercalating fluorescent dyes, GelRed (red), DAPI (blue), and SYBR green I (green). Under UV irradiation, the bioplastic building blocks exhibit uniform coloration (Fig. 3c i, d i), indicating homogeneous DNA distribution. They are assembled using the aqua-healing method, resulting in well-formed centimeter scale products with slightly discolored appearances (Fig. 3c ii, d ii), but vibrant, high-contrast fluorescence under UV irradiation (Fig. 3c iii, d iii), validating the building block strategy. Additionally, the assembled bioplastics exhibit excellent durability. A 3D flower-shaped sample retains its shape and high-contrast fluorescent colors (Fig. 3d iv) after 2 months of storage in Singapore's challenging environment (23-34 °C, 80-90% humidity), which is harsher than cold or arid climates.

Moreover, such bioplastics offer high-precision manufacturing down to the nanoscale. This capability is demonstrated through three representative applications, where plastics are commonly used, including biomimetic surfaces (Fig. 3e), medical microneedle patches (Fig. 3f), and photonic nano-patterns (Fig. 3g). A direct drying method with specific templates was applied to achieve precise nanoscale fabrication. The biomimetic surface accurately replicates the original surface texture of a Lotus leaf (Fig. 3e). The microneedle patches feature well-defined conical structures with ~300 μm in diameter and sharp tips (Fig. 3f), and can penetrate the stratum corneum of mouse (Fig. S22), which might be further developed as monitoring and delivery devices. The nano-patterns are commonly used for photonic devices, to which the bioplastics are also applicable. The bioplastics are produced into a series of patterns, including rectangular textures (Fig. 3g i), grids (Fig. 3g ii), and pyramid arrays (Fig. 3g iii). Remarkably, no defects are observed across tens of micrometers (Fig. 3g i), and the feature size reaches as fine as ~150 nm (Fig. 3g ii and iii). These results highlight the promise of bioplastics for precise, durable products across multiple scales.

To assess its biodegradability, we conducted an experiment in a practical case using a spoon-shaped bioplastic sample alongside a commercial PE-based disposable spoon as a reference. Both were buried in local soil. Over time, the bioplastic sample decayed gradually, achieving complete degradation after 29 days, while the reference PE spoon retained its original shape throughout the experiment (Fig. 3h). We also conducted a biodegradation test according to the standards of American Society for Testing and Materials (ASTM), and demonstrated that the degradation ratio is ~28% after 45 days (Fig. S23). Moreover, we demonstrated that the degradation process can be accelerated with nucleases. The bioplastic sample, stained with GelRed for better illustration, was monitored under UV irradiation to visualize the degradation process. In water containing DNase I nuclease, the red bioplastic gradually decreased in size and fully disappeared in 120 minutes (Fig. 3i), significantly faster than in soil. Increasing the nuclease concentration further reduced degradation time to 10-20 minutes (Fig. S24a). The rapid biodegradation in the lab is attributed to the DNA content in the bioplastic, which makes it susceptible to enzymatic breakdown. In contrast, without nuclease, the red-stained bioplastic remained intact in water after 120 minutes (Fig. S24b). Incorporating DNA into the bioplastic matrix offers a key

advantage by introducing an additional, orthogonal degradation pathway beyond those available to polysaccharide-based systems alone. While enzymatic degradation of polysaccharides—such as dextran hydrolysis by dextranase—is well-established, these pathways are typically limited to specific glycosidic linkages. DNA, in contrast, is easily degraded by a distinct set of enzymes (e.g., DNase), which are abundant in many natural environments. This approach not only broadens the environmental triggers for material breakdown but also enables more tunable and potentially faster degradation profiles. Moreover, the inclusion of DNA adds chemical diversity to the bioplastic, offering additional handles for molecular design and programmability that are not easily accessible with polysaccharides alone. These results highlight minimal environmental risks posed by uncollectable bioplastic waste, e.g., the threat of microplastics to the ecosystem, and suggest that enzymatic degradation could serve as an efficient end-of-life treatment for large-scale bioplastic disposal in industrial applications.

To demonstrate the universality of this method to produce polysaccharide-DNA bioplastics, we replicated the freeze-drying process by replacing Dex with either AA (Fig. 4a–c) or CMC (Fig. 4d–f). Both AA and CMC were oxidized using $NaIO_4$, with their respective oxidization mechanisms illustrated in Fig. 4a, d, respectively. Both the AA-DNA and CMC-DNA bioplastics are recyclable and reusable through the same water-processable method. The typical recycling process was demonstrated using samples with 5% DNA and 10% polysaccharides (AA or CMC), as illustrated in Fig. 4b, e. It is demonstrated that bioplastics could be produced using hydrogels with a broad range of AA or CMC concentrations (2- 20 wt.%, Fig. 4c, f). As the AA or CMC concentrations increased from 2 to 20 wt.%, both AA-DNA and CMC-DNA hydrogels exhibited an increasing G′ value (Fig. S25). The resulting bioplastics were found to be recyclable through water-based processing (Fig. S26a-b and S27a-b), and both types also demonstrated aqua-healing capabilities (Fig. S26c and S27c). Additionally, mechanical tests revealed that Young's modulus averaged ~116 MPa for 10%AA-5%DNA sample and ~92 MPa for 10%CMC-5%DNA sample (Fig. S28), highlighting their robustness and versatility.

To assess the biocompatibility of the bioplastics, we conducted in vitro cytotoxicity tests using primary human dermal fibroblasts and in vivo digestion experiments on mice. The in vitro assessments suggest that cell viability remains above 80% across all three bioplastics: Dex-DNA, AA-DNA, and CMC-DNA (Fig. 4g) within a concentration range of 0 to 12.5 mg/mL, with no significant declining trends observed. For the in vivo evaluation, histological examination demonstrated no noticeable differences in body weight between mice fed with the three bioplastics (Dex-DNA, AA-DNA, and CMC-DNA) and the control group after 14 days (Fig. S29a). Moreover, no observable pathological damage was found in major organs, including heart, intestine, kidney, liver, spleen, and stomach, when compared with the control group (Figs. 4h, i, and S29b). Notably, unlike microplastic residues that pose health risks to organisms[12–14], no such residues were detected in the organ sections. These results indicate the biocompatibility and biosafety of the polysaccharide-DNA bioplastics, which can be attributed to the inherently safe nature of their raw components —polysaccharide and biomass-derived DNA.

We further demonstrate the scalability of the production process and assess the potential environmental impacts of upscaling the production of these bioplastics, using the 10%Dex-5%DNA formulation as a case study. Large-scale hydrogel batches of 500 mL were successfully produced (Fig. 5a), and the bioplastic foils covering ~12.5 ×12.5 cm² were fabricated (Fig. 5b), alongside various morphologies designed to meet potential commercial demands (Fig. 5c). Interestingly, the bioplastic foils with a high solid content exhibit a transparent appearance (Fig. 5b), contrasting with the white, opaque appearance of products generated through freeze-drying (Fig. 5c). This difference arises from the microstructural features, where the porous structure of

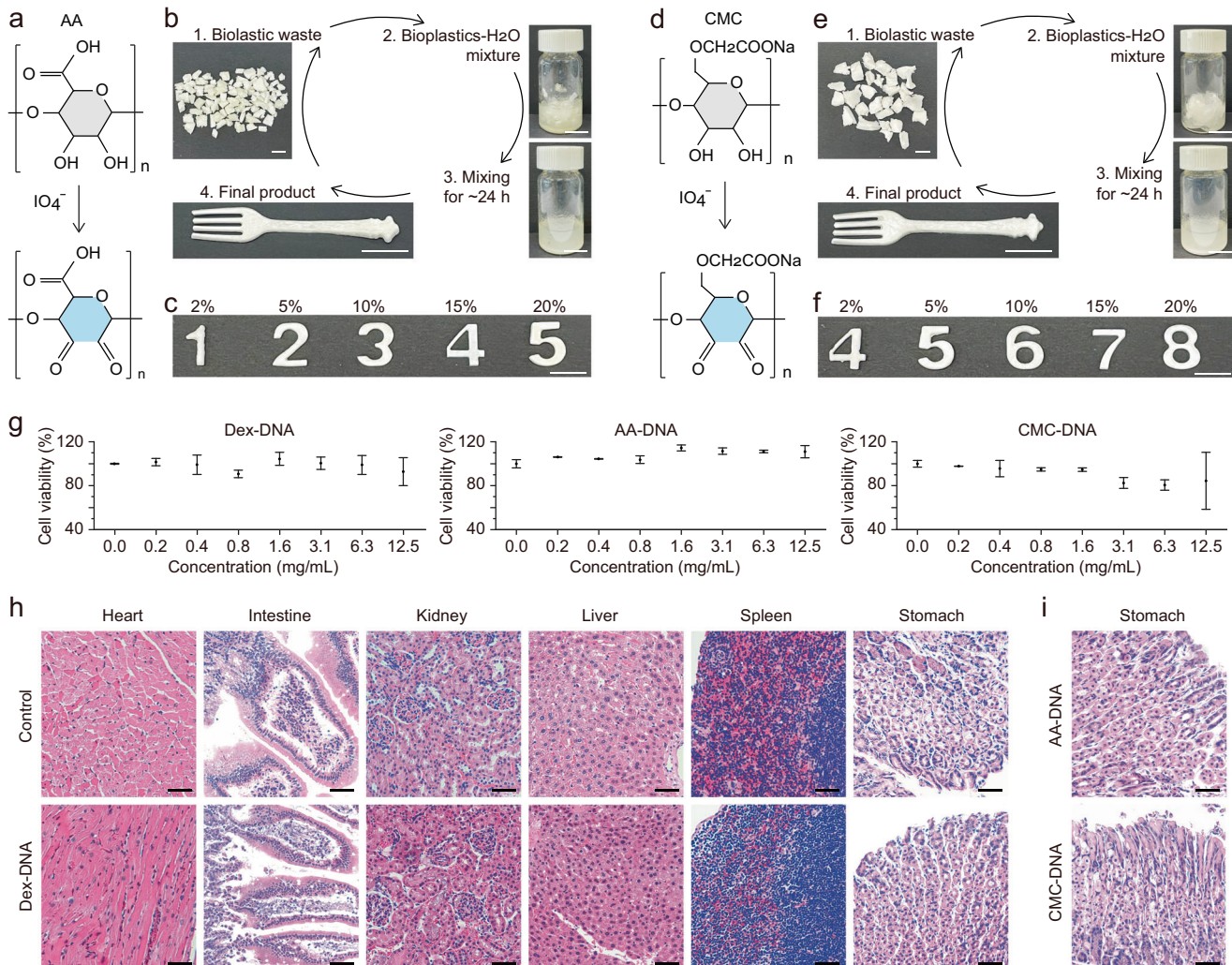

**Fig. 4 | Universality of the method and bioplastic biocompatibility.** Molecular structure changes of (**a**) AA and (**d**) CMC under an oxidization process. Illustration of recyclable processes of the (**b**) AA-DNA and (**e**) CMC-DNA bioplastics. Photographs of the (**c**) AA-5%DNA and (**f**) CMC-5%DNA bioplastics with different polysaccharide contents of 2%–20%. **g** Cytotoxicity assessment of the three bioplastic formulations, showing cell viability across concentrations. Data are presented as mean±standard deviation (n = 3). **h** Representative microscopy images of stained mice organ sections 14 days after ingesting Dex-DNA bioplastics, with comparisons to control group images. Examined organs include heart, intestine, kidney, liver, spleen, and stomach. **i** Representative microscopy images of stained mice stomach sections 14 days after exposure to AA-DNA and CMC-DNA bioplastics. The scale bar is 1 cm in (**a**–**f**) and 50 μm in (**h**, **i**).

freeze-dried samples scatters visible light more strongly, resulting in the white appearance. These results indicate that the optical properties of the bioplastics can be customized to suit various practical applications, offering flexibility for product design and functionality.

To assess the environmental impacts of the produced bioplastics, we conducted a cradle-to-grave *ex-ante* LCA[38–40], and compared them with widely used commercial plastics, including PS, polyethylene terephthalate (PET), polyvinyl chloride (PVC), LDPE, high-density polyethylene (HDPE), polypropylene (PP), and polylactic acid (PLA) (Fig. 5 d, e). The functional unit for the assessment is 1 m³ of plastic for single-use applications. The end-of-life scenario assumes industrial composting for both the bioplastic and PLA, while landfilling is considered for other commercial plastics. The detailed recycling process of the bioplastics is documented in the Methods section. When the bioplastic is recycled 10 times, its life-cycle Global Warming Potential (GWP) is 861 kg $CO_2$-eq m$^{-3}$, 58.7% to 78.3% lower than that of commercial plastics without recycling (Fig. 5d). In contrast, without recycling, the GWP of the bioplastic rises to 6,332 kg $CO_2$-eq m$^{-3}$, 59.4% to 203.3% higher than the commercial plastics. This is due to emissions from the production phase are distributed across multiple reuse cycles

when recycling cycles increase (Fig. 5d). Given the significant impact of electricity on the GWP, we analyze how different regional electricity sources influence the results in various geospatial locations near Singapore (Fig. 5e). Along with Singapore, Myanmar, Japan, Vietnam, and South Korea exhibit relatively low GWP values (832–1002 kg $CO_2$-eq m$^{-3}$). In contrast, regions with more carbon-intensive energy grids, such as India, Indonesia, Philippines, and China, display higher GWP values (1,186-1,560 kg $CO_2$-eq m$^{-3}$). These results suggest the importance of adopting low-carbon electricity for bioplastic production and recycling. The comparison results to commercial plastics that are recycled 10 times are also available in Fig. S30. The results of other environmental impact categories are available in Fig. S31 with the details of the contribution analysis in Table S1. Given that the current production process is still at the lab scale, further reductions in life-cycle greenhouse gas emissions are expected through process optimization and sustainable raw material sourcing in large-scale manufacturing. While offering environmental benefits, the current lab-scale production of these bioplastics incurs a raw materials cost of 3.1-5.2 USD/gram (Table S2), which is relatively high. Future cost reduction, including more affordable extraction techniques, optimization of raw

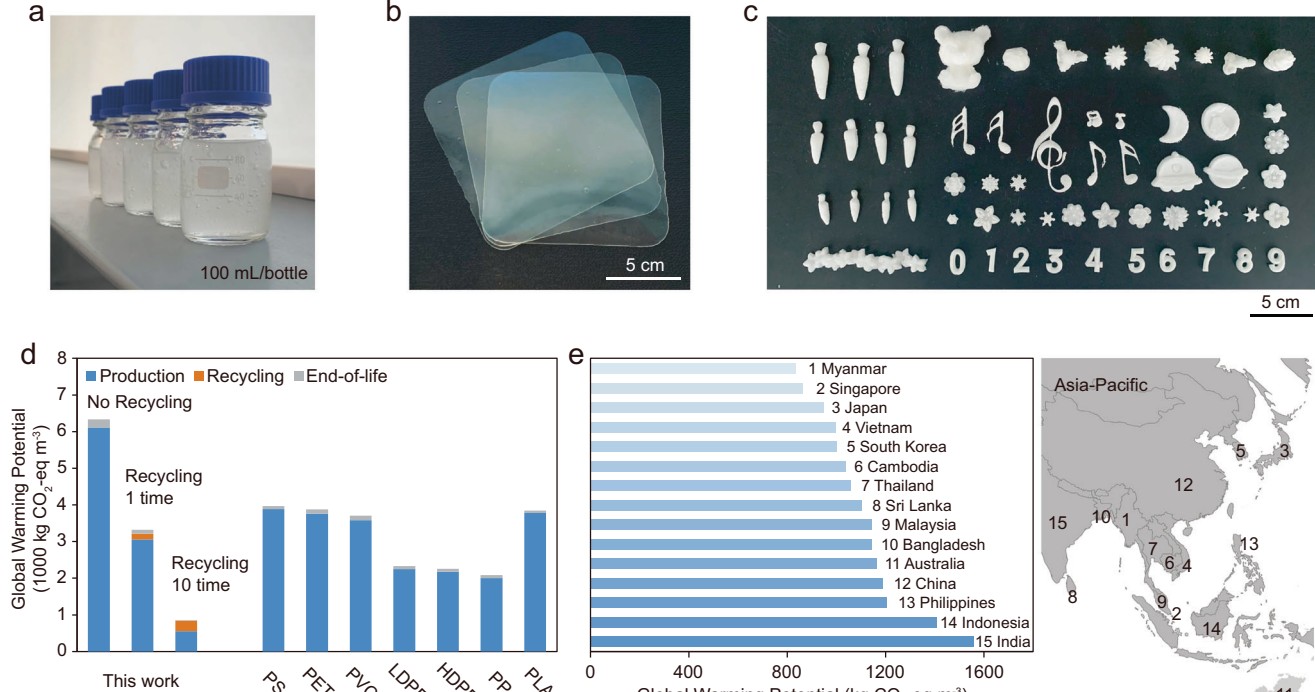

**Fig. 5 | Production upscaling and environmental impact analysis. a** Photographs of the upscaling production of 10%Dex-5%DNA hydrogels. Images of bioplastic products in foil form (**b**) and with various morphologies (**c**), created using 10%Dex-5%DNA hydrogels. (**d**) Comparison of the Global Warming Potential (GWP) per use cycle across different recycling times for the bioplastics versus commercial plastics, including polystyrene (PS), polyethylene terephthalate (PET), polyvinyl chloride (PVC), low-density polyethylene (LDPE), high-density polyethylene (HDPE), polypropylene (PP), and polylactic acid (PLA). **e** GWP of bioplastics recycled 10 times across different countries/regions. Map adapted from a public domain image on Wikimedia Commons (https://commons.wikimedia.org/wiki/File:Location_Map_Asia.svg). No copyright restrictions apply.

material supply chains, and the use of more energy-efficient fabrication equipment, are anticipated to be critical factors for successful commercialization.

In summary, the present study introduces a sustainable bioplastic material, revealing multi-closed-loop recyclability and water (re)processability. The bioplastics are based on abundant polysaccharide biomass products, such as Dex, AA, CMC, and DNA of plants or living organism waste. Chemical (NaIO₄), partial oxidation of the polysaccharides to aldehyde−functionalized polysaccharides followed by reversible imine covalent bond formation with amine functionalities associated with DNA yield water processable polysaccharide/DNA crosslinked hydrogels as a raw material to produce sustainable bioplastic products. The plastic products exhibit (bio)degradability and recyclability reflected by the aqueous recycling of the parent gel constituents by hydrolysis of the plastics and biodegradability of the DNA/polysaccharides by native biocatalysts (enzymes), e.g., bacteria. The resulting bioplastics reveal resistance towards organic solvents, aqua-healing capacities, scalability, and effective processing across nanometer, micrometer, centimeter to meter scales, suggesting broad and versatile applications. The LCA reveals that the production of 1m³ of the bioplastic results in 861 kg $CO_2$-eq m⁻³ emission when recycled 10 times, which is 58.7%-78.3% lower than most commercial plastics without recycling. Moreover, we note that the DNA extracted from different sources might differ in sequence, length, and secondary structure. These might affect the resulting bioplastic due to differences in the imine bond densities, and consequently, influence their mechanical properties (e.g., Young's modulus). Moreover, the key characteristics of the bioplastics, including enzymatic degradability and close-loop recyclability, should be retained. Nevertheless, harnessing pure sequence-programmed DNA stocks could be produced by genetically-engineered crops. While such approaches are at present cost-prohibitive and impractical for large-scale sustainable material

production, for special material applications, adaptation of these concepts might be interesting. Furthermore, the concept of polysaccharide/DNA bioplastics based on reversible imine-crosslinked frameworks may be extended to polysaccharide/protein crosslinked matrices. At present, the need for a cost-effective "clean" source for NaIO₄ is a limiting factor in the process. The use of solar light to produce IO₄⁻ or, alternatively, the solar-light photocatalyzed oxidation of the polysaccharides to the aldehyde-modified polysaccharides could potentially overcome this limitation. Photocatalyzed oxidation of iodide to periodate in aqueous basic solutions, or photocatalyzed oxygen-mediated Fenton oxidation of I⁻, could provide parallel close-hoops for the present process, synthesizing the bioplastic materials. The present study provides, however, important potential pathways, advancing sustainable, environment-friendly bioplastic materials.

## Methods
### Materials
The deoxyribonucleic acid sodium salt from salmon testes (DNA, Sigma Aldrich), dextran from Leuconostoc spp. (Dex, Sigma Aldrich, Mr. ~40,000), alginic acid sodium salt from brown algae (AA, Sigma Aldrich), carboxymethyl cellulose (CMC, Sigma Aldrich, Mw ~250,000), ethylene glycol (J.T.Baker, ≥99.0%), sodium periodate (NaIO₄, Sigma Aldrich, ≥99.8%), hydroxylamine hydrochloride (NH₂OH·HCl, Sigma Aldrich, 99.999%), sodium hydroxide (NaOH, Sigma Aldrich, ≥98.0%), GelRed@Nucleic (Biotium, 10,000 x in water), SYBR Green I (Lonza, 10,000x in DMSO), 4',6-diamidino-2-phenylindole (DAPI, ThermoFisher Scientific, 1 mg/1uL), DNase I (ThermoFisher Scientific, RNase-free, 1 U/μL), n-hexane (Sigma Aldrich, >95.0%), chloroform (Sigma Aldrich, >99.0%), dimethyl sulfoxide (DMSO, Sigma Aldrich, >99.5%), polyethylene (PE, Sigma Aldrich), and polystyrene (PS, Sigma Aldrich) were used as received. Oligomers (A9, C9, G9) were purchased from Integrated DNA Technologies PTE, Ltd.

Distilled water (DI water, 18.2 ΩM·cm) through a Mili-Q purification system (i-Drop-VF-T, Arium Pro) was used.

## Synthesis of oxidized polysaccharides

The polysaccharides (Dex, CMC, and AA) were oxidized using $NaIO_4$ following the same procedure. Specifically, 1.0 g of each poly-saccharide was dissolved in 8.0 mL DI water, followed by the dropwise addition of $NaIO_4$ solution. The $NaIO_4$ solution was prepared by dissolving 0.264 g of $NaIO_4$ in 2.0 mL DI water. The oxidization reaction proceeded under continuous stirring for 6 hours at room temperature and was terminated by adding 10 mL of ethylene glycol. The oxidized polysaccharides were purified three times by using DI water with an Amicon filter (MWCO of 10 K Da). Centrifugation at 10414×g for 30-60 minutes was employed to facilitate the purification. The residue was collected by dissolving it in DI water, followed by freeze-drying to obtain the solid oxidized polysaccharides. The oxidization degree for all three polysaccharides was determined to be 40-50% using a standard hydroxylamine hydrochloride reaction[41,42]. Briefly, 0.1 g of the oxidized polysaccharides was dissolved in $NH_2OH \cdot HCl$ solution (25 mL, 0.25 M) for 2 hours at room temperature, which was then titrated against NaOH (0.087 M).

## Preparation of bioplastics

The DNA was dissolved in DI water at a fixed content of 5 wt.% through continuous shaking overnight. Oxidized polysaccharides (at 2 wt.%, 5 wt.%, 10 wt.%, 15 wt.%, and 20 wt.%) were then added, and the mixture was stirred, followed by overnight shaking to form hydrogels. Solid bioplastic samples were obtained through a drying process, with silicone mold used to shape the final products. Unless otherwise specified, a standard freeze-drying procedure at −20 °C was applied to sample preparation. Dyed samples were prepared by mixing 2.0 mL of DNA hydrogels (5 wt.%) with 2.5 μL dyes, including GelRed (red), SYBR Green I (green), and DAPI (blue). The solid content in the hydrogels (wt.%) is used to label the samples unless otherwise specified.

## Micro- and nano-structure control

The microporous bioplastics were prepared using a freeze-drying process, either in a −20 °C freezer or in liquid nitrogen at −196 °C. For a given solid content, rapid freezing with liquid nitrogen was conducted to produce bioplastics with small micropore sizes. Non-porous bio-plastics were produced by air-drying hydrogels with a 2% solid content at room temperature for approximately three days. This method was also applied to create precise micro- and nano-structured surfaces. A pristine lotus leaf was used as a template for the biomimetic surface replication. Silicone- or silicon-based molds were used to fabricate nanoneedle arrays or photonic nano-pattern surfaces, respectively.

## Bioplastic recycling

Bioplastic waste was mixed with DI water to match the original content used in the initial formulations. The mixture was vigorously shaken for approximately 24 hours, producing soft hydrogels for high solid content or viscous liquid for low solid content. These hydrogels or liquids were then processed using the same molding and freeze-drying methods as the original preparation to generate recycled bioplastic products. For recycling mixed plastics, the total plastic content was kept below 2 wt.%. The recycling process involved selectively dissolving bioplastics in water and PS in chloroform. The liquid and sediment were separated by filtration, and the recovered liquid phase was air-dried at room temperature to get the recycled solid plastics.

## Aqua-healing and chemical resistance tests

Polysaccharide-DNA bioplastic samples (10 wt.%-5 wt.%) were used for this experiment. For fracture interface testing, the broken surface was lightly wetted with DI water and re-connected after removing the water

at 60 °C. For the chemical resistance test, bioplastic samples were produced in $1 \times 1\, cm^2$ square shapes and placed in 20 mL glass bottles or petri dishes containing n-hexane, chloroform, DMSO, ethylene glycol, glycerol, or formamide. Top-down photographs were taken to document the results.

## Biodegradation test

An ASTM standard D6400-23 was applied for the biodegradation test for 45 days. The degradation level was assessed by the amount of released carbon dioxide, which was measured through a titration method to test the quantity of dissolved carbon in sodium hydroxide absorption solution. The biodegradation of DNA was also conducted using DNase I. The dyed bioplastics, 100 mg, 0.021 mL of DNase I, 0.1 mL of $MgCl_2$ buffer, and 0.879 mL of DI water were mixed and incubated at 37 °C. A non-enzyme control test was conducted without DNase I, while an accelerated biodegradation test was conducted using 0.418 mL DNase I. Photographs of the dyed samples were captured under UV irradiation to monitor the degradation process. An outdoor biodegradation test was conducted in Singapore with a daily temperature range of 23-34 °C and humidity levels of 80-90%. The samples were buried in soil during the test and unearthed periodically for photographic documentation.

## In vitro cytotoxicity test

Evaluation of the cytotoxic effects of DNA bioplastics was examined by applying a CellTiter-Glo® 2.0 Cell Viability Assay (Promega)[43]. Primary human dermal fibroblasts (HDF, PCS-201-012, ATCC) were cultured in DMEM supplemented with 10% fetal bovine serum (Thermo Fisher Scientific) and 1% penicillin/streptomycin (Thermo Fisher Scientific) and maintained at 37 °C in an incubator containing 5% $CO_2$. Cells were allowed to reach 80% to 90% confluence before being subcultured with 0.05% trypsin-ethylenediaminetetraacetic acid. The 96-well plates with each well containing ~10,000 cells were cultured for 24 h. Then, the existing media in the wells were replaced with 100 μL of the diluted bioplastic solutions, followed by an additional 24-hour incubation. The diluted bioplastic solution was prepared by dissolving the bioplastics (10%Dex-5%DNA, 10%AA-5%DNA, and 10%CMC-5%DNA) in the medium at concentrations ranging from 0.05 to 12.50 mg/mL. A 100 μL of assay reagent was introduced to each well for cell lysis. The plates were then left at room temperature for 10 minutes to stabilize the luminescent signal. Luminescence readings were measured with a microplate reader (Tecan Safire), and cell viability in percentage was calculated in comparison to the control cells.

## In vivo test

All animal procedures received approval from the Institutional Animal Care and Use Committee, Biological Resource Center, Singapore. Female BALB/c mice (18-22 g) between 7 and 8 weeks of age were obtained from InVivos Pte Ltd, Singapore. The mice were kept at 22 °C with 50% humidity and on a 12-hour light/dark cycle with access to food and water *ad libitum*. They were randomly divided into two groups of five: a control group and a bioplastic treatment group. All three-bioplastic treatments of 10%Dex-5%DNA, 10%AA-5%DNA, and 10%CMC-5%DNA were conducted in the same method. Each bioplastic was dissolved in DI water at a concentration of 1000 mg/mL and given orally at a dosage of 10 g/kg body weight. The control mice received an equivalent volume of DI water. The animals were examined every 30 minutes during the first 2 hours post-treatment and then monitored daily for 14 consecutive days. Body weights of the mice were measured before treatment and at the end of the observation period. The main organs were collected after 14 days of sample administration and fixed in 10% neutral formalin and embedded in paraffin for histology analysis. The embedded samples were cut into 4 μm thick sections and stained with hematoxylin and eosin (H&E) for photographing under an optical microscope.

## Microneedle penetration test

Animal studies were conducted to test the penetration of micro-needles into the stratum corneum. Female C57BL/6 mice (18-22 g) between 8 and 9 weeks of age were obtained from InVivos Pte Ltd, Singapore. The mice were kept at 22 °C with 50% humidity and on a 12-h light/dark cycle with access to food and water *ad libitum*. The fur at the microneedle application site on the back (approximately $2 \times 2$ cm$^2$) was removed under anesthesia one day prior to sample application. The microneedle was applied topically with gentle pressure. After removal, photographs of the application site were taken. Skin samples from the microneedle access sites were collected, fixed in 10% neutral-buffered formalin, and embedded in paraffin for histological analysis. The paraffin-embedded tissues were sectioned at 4 μm thickness and stained with H&E. The stained slides were examined and photographed using an optical microscope.

## WVTR and permeability (*P*) tests

The WVTR and *P* of prepared bioplastics were carried out by sealing circular thin film on vials containing DI water. The initial weights of the vials were recorded, and the vials were placed in a humidity chamber (38 °C, 90% RH, Memmert HPP110eco). After 24 hours, the vials were reweighed to determine the water vapor loss through the bioplastic films, WVTR was calculated according to Eq. 1,

$$\text{WVTR} = \frac{\Delta W}{A \times t} \tag{1}$$

where $\Delta W$ is the loss of weight in gram, $A$ is the cross-sectional area (m$^2$) and $t$ is time in day.

*P* was calculated according to Eq. 2,

$$P = \frac{WVTR \times L}{\Delta p} \tag{2}$$

where $L$ is the thickness of the film (m), $\Delta p$ is the partial pressure difference ($\Delta p = p_{sat}(38°C) \times (RH_1 - RH_2)$).

## Characterizations

The rheological measurements were performed using a Rheometer (Thermo Scientific HAAK™ MARS™) with a parallel-plate geometry (35 mm in diameter) at room or elevated temperature. The micro/nanostructure was imaged using a field scanning electron microscope (JEOL, JSM-7400F) at an accelerated voltage of 5 kV, and the sample was coated by Au for 30 s using a sputter coater (JEOL, JFC-1600 AFC) before measurement to promote sample conductivity. The mechanical property of produced bioplastics was measured through a tensile test using Mechanical Tester MTS Model C4. Fluorescent photographs were taken using an iPhone X under the irradiation of a UV lamp (Viber, VL-6.LC, 6 W). As to the NMR test, samples were dissolved in D$_2$O, and $^1$H-NMR spectra were recorded on Bruker AV400SB (400 MHz, ns = 178). Chemical shift (δ) is reported in parts per million (ppm) relative to the residual solvent peak of water (δ = 4.8). The FTIR spectra were recorded using an FTIR spectroscopy with attenuated total reflection (ATR) attachment (Bruker Vertex 80 v) at a resolution of 1 cm$^{-1}$ with 64 scans per sample. Differential scanning calorimetry (DSC) data were collected using DSC Q100 (TA instruments) under nitrogen. A scan rate of 20 °C/min was employed in the temperature range of −40 °C to 200 °C. All statistical analyses are conducted on three samples. Data was analyzed and graphed by OriginPro 2017, Adobe Illustrator 2024, etc.

## Life Cycle Assessment

The LCA in this study followed the ISO standard series 14040 by using the software OpenLCA 2.0. The selected functional unit is 1 m$^3$ of bioplastics (150 kg) used for 1 time. The density of the bioplastic is set to be 150 kg m$^{-3}$, close to the bioplastic's density in the experiment (15 wt.% solid polymers). The feedstock for DNA extraction is assumed to be waste plant leaves, which reflects a plausible, low-impact biomass source for early assessment purposes. This may evolve with future supply chain development. Since the bioplastic density is lower than the current commercial plastics, given the same functional shape, it is reasonable to compare on the volume basis instead of the mass basis in this study. The system boundary is *ex-ante* cradle-to-grave, including raw material extraction, transportation, production, and end-of-life. The upstream burdens of producing materials and energy products used in the system are included. The end-of-life scenario for bioplastic in this study and PLA is assumed to be industrial composting, while landfilling is assumed for other commercialized plastics. The environmental impact assessment method adopted TRACI 2.1 developed by U.S. Environmental Protection Agency with Global Warming Potential 100 years factors recently released by the Intergovernmental Panel on Climate Change. See details in the Supporting Information (Note S1 and Table S3 and S4), including the recycling process. It is worth noting that the recycling process may be influenced by various conditions (e.g., contamination, UV exposure, and environmental stressors) that can lead to more operations. The inventory data of the LCA were majorly upscaled based on the experimental data that can be further improved or optimized along the commercialization as the investigated technologies were at the early stage[40]. The purpose of our cradle-to-grave *ex-ante* LCA is to provide a preliminary baseline assessment of the environmental performance of the proposed bioplastic compared to conventional plastics. This early-stage analysis serves as a directional guide for process optimization and design improvement, rather than a definitive environmental impact statement.

## Reporting summary

Further information on research design is available in the Nature Portfolio Reporting Summary linked to this article.

## Data availability

The data generated in this study are provided in the main text and Supplementary information. Data is available from the corresponding authors on request. The $^1$H NMR data are available at https://doi.org/10.5281/zenodo.15878478. Other Source data are provided with this paper.

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

## Acknowledgements

The research is supported by A*STAR (CDF, C210112014, to Y.H.), and Science and Engineering Research Council (CRF, UIBR, KIMR220901A-sercrf, to Y.H.), Singapore. Y.K. acknowledges the support of start-up fund, faculty research grant (SISFRG2513), and the Lam Woo Research Fund (LWP20039) from Lingnan University, Hong Kong. Z.G.D. would like to acknowledge the funding support from National Research Foundation via Quantum Engineering Programme 2.0 (Award No. NRF2021-QEP2-03-P09).

## Author contributions

Y.K. designed and performed most of the experiments, data analysis and wrote the manuscript. K.L. performed LCA studies. J.W. participated in materials synthesis. H.L. and S.G. conducted in vitro and in vivo studies. K.R. and Z.L.D. performed mechanical studies. F.C. conducted NMR studies. W.L. participated in materials characterizations. J.L. commented on the materials properties. Z.G.D. participated in the micropatterns preparation. X.C. and I.W. mentored and commented on the study and revised the manuscript. Y.H. conceived and supervised the research, secured the funding and revised the manuscript. All authors reviewed and commented on the manuscript.

## Competing interests

All authors declare no competing interests.
