## [Transparent Peer Review file · Nature Communications]

Sustainable DNA-Polysaccharide Hydrogels as Recyclable Bioplastics

Corresponding Author: Professor Itamar Willner

Version 0:

Reviewer comments:

Reviewer #1

(Remarks to the Author)

Review of "Sustainable DNA-Polysaccharide Hydrogels as Reprocessable Bioplastics"

Overall Assessment:

One of the first and most obvious problems is the title: it has a key word "reprocessable" which readers will take as the most dominant feature of the reported bioplastics. However, throughout the entire writings, the word "reprocessable" occurs ONLY once, in the figure 1 legend. In other words, the main text of the whole paper never mentions the word "reprocessable" -- then why it is in the title?!

The manuscript presents an interesting approach to developing sustainable bioplastics by leveraging the reversible imine bonding between oxidized polysaccharides (dextran, alginate, and carboxymethyl cellulose) and biomass-derived DNA. The authors demonstrate multi-closed-loop recyclability, water-based processability, biodegradability, and scalability, along with a life cycle assessment (LCA). The experimental data, including mechanical properties, biocompatibility, and environmental impact analysis, are in great amounts, well-documented, and most solid.

However, several aspects require correction, clarification, additional experimentation, and refinement to strengthen the claims and enhance the manuscript's scholarly impact. The innovative aspect also needs to be elaborated more. Below, specific concerns and comments are provided; they need to be addressed before the work can be published.

Major Concerns and Comments

1. Introduction, Rigor, and Novelty

- **Comment on Reversible Bonding and Polymer Breakdown:** The authors write in the introduction: "formulas without reversible bonding features limit their potential in the breakdown of polymer chains, a critical factor for chemical recycling."

Polymer chain breakdown depends not only on reversible bonds but also on inherent degradability. This claim overstates the importance of reversible bonds and oversimplifies the complexity of polymer degradation. The authors should acknowledge that while reversible bonds (e.g., imine) facilitate chemical recycling, other factors like chain composition and environmental conditions (such as enzymes for bioplastics) also play significant, and might even be more important, roles. In addition, clarify how imine bonds specifically enhance recyclability compared to non-reversible bonds (both pros and cons).

- **Eco-Friendliness Claim:** In the authors' critique of prior DNA-based bioplastics (e.g., ref 24) as "not eco-friendly" ("...the chemical synthesis process is not eco-friendly, involving mass usage of organic solvents and relatively harsh synthesis conditions, e.g. a relatively high temperature under an inert gas environment"), the word of "chemical synthesis process" is ambiguous at least; if the ref24 is the target, then the statement is wrong. Please note that the authors' usage of sodium periodate (a hazardous chemical) in their process isn't entirely free of environmental concerns either. Thus, a blanket of assessment of "not eco-friendly" is not appropriate here. It is suggested that the authors should soften this critique, and explicitly and fairly compare their method with that of ref 24 (e.g., solvent use, temperature, energy input, etc.).

- Similarity wording (“unintentional plagiarism”?) to Ref 24: The sentence about biomass DNA availability mirrors ref 24 too closely without acknowledgement. It requires either a citation of ref24, or rephrasing to avoid unintentional overlap.

...DNA on the earth is estimated at 50 billion metric tons, and less than 1% of this biomass DNA reserves is more than enough to meet the annual demand on the feedstock of worldwide commodity plastics production

...DNA on earth is estimated to be 50 billion metric tonnes; less than 1% of it would be more than sufficient to fulfil the annual demand on the feedstock of global commodity plastic production.

- Novelty The author’s work is clearly a nice extension/addition of a previous reported work (ref24) with the incorporation of polysaccharides into bioplastics using a different, imine-bond crosslinking method. Please note that the polysaccharide-based bioplastics have been extensively reported for many years, thus, the novelty of this work over ref 24 (PEGDA-crosslinked DNA plastics) and other polysaccharides-based plastics needs sharper delineation and more extensive elaboration — e.g., emphasize the water-based process, more contents than DNA alone, better properties than polysaccharides, and potential synergy between DNA and polysaccharides --- to showcase DNA-polysaccharides’ distinct advantages.

2. Chemistry and Characterization

- Oxidation Reaction Misnomer: The periodate oxidation specifically refers to Malaprade reaction (L. Malaprade. Action of polyalcohols on periodic acid. Bull. Soc. Chim. France. 1928, 43: 683.; L. Malaprade. Oxidation of some polyalcohols by periodic acid-applications. Compt. Rend. 1928, 186: 382.). Criegee oxidation usually refers to the oxidation of vicinal diol using lead tetraacetate as the oxidant. The authors should correct this throughout the manuscript (e.g., Fig. 1, Methods, etc.) to ensure accuracy. The Malaprade reaction cleaves a carbon-carbon bond of vicinal diol structure, dextran or other polysaccharides are not exceptional. In Fig. 1, it looks like that the authors presented rearrangement forms of oxidized dextran, that should be improved and/or clarified as well.

- Imine Bond Evidence: The NMR evidence (Fig. S1) for imine bond formation is promising but insufficiently conclusive. Why not IR spectroscopy? IR could confirm the C=N stretch and provide evidence about hydrogen bonding interaction between DNA and polysaccharides. Additionally, the amine group from A, T, G, C of DNA has different nucleophilic activity. Usually, the amine group from G has the highest reactivity, but T definitely cannot react with aldehyde to form imine bond because the nitrogen atom in organic compounds can form a maximum of three valence bond according to valence bond theory. This should be corrected in Fig. 1. The authors speculate multiple reactive sites (e.g., N2 of guanine, N6 of adenine) but provide no confirmation. It needs to confirm/clarify which base reacted with the aldehyde on oxidized polysaccharides.

3. Water Processability and Resilience

- Solvent-Free Claim: The authors claim “solvent-free water (re)processability,” but water is a solvent; thus, the claim of solvent-free is misleading. It should be deleted.
- Water Resilience Concern: Water processability is a double-edged sword: it implies poor moisture resistance, a known challenge for bioplastics. The authors tout this as an innovation/advantage, but applications requiring water stability (e.g., packaging) may be limited. This point should be discussed (pros and cons), and overselling should be avoided.

4. Mechanical Properties and Recycling

- Imine Bond Stability Post-Recycling: The mechanical properties remain stable after 10 cycles (Fig. 2f), which is impressive. To directly link the reversible chemistry to recyclability, NMR/IR should be used to track imine bond integrity post-recycling. This could strengthen the multi-closed-loop claim.
- Solvent Resistance: The bioplastics resist hexane, chloroform, and DMSO (Fig. 3a) but dissolve in extreme pH (Fig. S6). What about the solubility in glycerol/ethylene glycol (Polar but viscous; may act as plasticizers), and formamide (highly polar)? which are other common relevant organic solvents. These testing are optional but if conducted, hopefully will broaden the resistance profile, adding more evidence to strengthen the resistance claim.

5. Life Cycle Assessment (LCA)

The authors present a cradle-to-grave LCA to evaluate the environmental impacts of DNA-polysaccharide bioplastics, which is commendable for its effort and aligning with sustainability goals, but it is quite premature for conclusive or even informative results because the reported bioplastics is still at a very early stage of the lab-scale: there is no large scale production, the entire process is not optimized, and the industry-scale parameters (e.g., energy efficiency, materials yields, specific recycling infrastructures, etc.) are all speculative. The authors should trim down significantly the entire LCA sections and emphasize the preliminary and speculative nature of this exercise and the non-conclusive characteristics of results (e.g., is DNA production really contributing to 99% Ozone depletion? Is this conclusion reliable/convincible?)

More specifically:

1. There exist huge data extrapolation risks when conducting cradle-to-grave LCA at this early stage.

2. Feedstocks are entirely unknown.
3. Recycling assumptions are also extremely speculative as there are no insights whatsoever at this stage how the recycling will be accomplished.
4. Lab scale recycling (shaking etc.) isn't the same as real-world recycling, which includes contamination, extreme and uncontrollable weather, UV, etc.

Recommendation

This is clearly a nice piece of research that pushes DNA-based materials broader and further. Data is solid and supports most of the conclusions (except the LCA work). Major revisions, mostly on writing, are needed; especially, some overstated claims need tempering.

Reviewer #2

(Remarks to the Author)

Reviewer #3

(Remarks to the Author)

The development of sustainable plastics is crucial for the sustainable development of human society. Although great efforts have been made and various sustainable plastics have been reported, the development of DNA-based bioplastics is still an exciting advance. In this paper, the authors demonstrate a facile way to produce bioplastics derived from biomass DNA and polysaccharides. These bioplastics exhibit attractive properties, such as water processability, biodegradability, good recyclability, scalability, high manufacturing precision, biocompatibility, etc., and also can minimize the risk of microplastic pollution and reduce carbon emissions. These results are impressive, and certainly of interest for the readers of Nature Communications. Therefore, I would like to recommend the publication of this manuscript after addressing the following concerns.

1. When competing with conventional plastics, the cost of the material is also a very important issue. It would be better to add a discussion of the potential cost of the biomass-based bioplastics in large-scale applications.
2. As the carrier of genetic information, DNA derived from different sources may have different sequences and structures. Would this affect the properties of the resulting bioplastics? And, is it possible to utilize the genetic information in the production and application of the DNA-based bioplastics?
3. The fact that the structure of the bioplastics can be affected by water also makes the application of these bioplastics very limited. Is it possible to find a way to solve this problem?
4. In Fig. 3f, the authors demonstrated the production of a medical microneedle patches with the bioplastics, which is very interesting. Could the microneedles penetrate the stratum corneum?
5. The authors demonstrated a sequential molding and drying process to form molded bioplastic products from hydrogels. A more detailed discussion of the possible molding mechanism is suggested.
6. There are some grammatical errors and typos in the manuscript, for example on page 2 line 58-59, "and and contributed significantly to carbon emissions".

Reviewer #4

(Remarks to the Author)

The article "Sustainable DNA-Polysaccharide Hydrogels as Reprocessable Bioplastics" describes the synthesis of a biodegradable hydrogel from dextra, alginic acid, carboxymethyl cellulose, and DNA. While the material seems innovative, the authors make a number of broad statements with little experimental evidence (see below). I would suggest major revisions before the article can be considered suitable for publication.

1. Paragraph 2 of the Results and discussion section reads more like an introduction or conclusion paragraph and should be removed given that no experiments are discussed to support the claims.
2. More material characterization should be provided, both of the original and recycled polymer (e.g., barrier properties, melting point). How do these properties link to the suggested applications of biomimetic surfaces, micro-needles, and photonic nano-patterns? Furthermore, how do the measured tensile properties compare to other bioplastics or conventional plastics?
3. Is there an ASTM standard that could be used for measuring biodegradability, rather than the relatively uncontrolled experiment presented in Fig. 3?

4. Language should be toned down. For example, a 500 mL batch is certainly up-scaling the process, but should not be considered "mass-producing".
5. The presentation of the LCA results in the abstract, introduction, and conclusion are misleading. Upon reading the results section, it seems that the 59-78% reduction in GHG emissions relative to conventional plastics is only achieved with the bioplastic is recycled 10 times. Please include this important information in the abstract, introduction, and conclusion. Also consider running a scenario in which conventional plastics such as PET or HDPE are mechanically recycled 10x.
6. What is the justification for using a volumetric functional unit for the LCA (m³) rather than the more common mass-based functional unit (kg or metric ton)?
7. Please check for typos throughout. For example, on page 4, "to form sold bioplastic products" should be "to form solid bioplastic products".

Version 1:

Reviewer comments:

Reviewer #1

(Remarks to the Author)

The authors have responded to my criticisms professionally and thoughtfully, and I am excited to recommend it for publication.

That being said, I continue to have a concern about the LCA, which should not affect my recommendation; rather, it serves as a documented record of my concern and also as a suggestion to further enhance the paper's rigor and impact.

The authors' argument that their LCA can provide a "preliminary baseline" for process optimization is noted, and I understand that the authors are trying to connect their new material closely and quantitatively with sustainability through LCA. However, the problems are:

1) The value of any LCA hinges on the reliability and process-relevance of the underlying data. At this stage, the authors' data lack the maturity to support meaningful environmental impact projections. Their statement that the technology is "...not subject to iterative optimization yet" further underscores this issue. Such ex-ante LCA will be very limited in its utility as a future "guide"; rather, premature LCA is more likely to risk producing false precision and misleading guidance.

2) While LCA can theoretically be applied to any new and old material/process, ex-ante LCA is most valuable as a dynamic tool within active process development where assumptions can be iteratively refined. However, once published in peer-reviewed literature, these preliminary, or worse, premature analyses become static references that may be cited without full appreciation of their underlying limitations, particularly as the foundational assumptions, including processing and recycling methods, become established and/or outdated.

3) The paper itself without LCA is good enough to be published – the material's synthesis, characterization, degradation, and applications are all quite impactful. However, adding this ex-ante LCA raises questions, reduces readers' focus, and diminishes the paper's rigor.

Suggestions (my personal preference, no need to follow):

remove LCA from the abstract. In the discussion section, discuss the sustainability/recycling aspects of the material and only include some of the LCA data in the supplementary materials."

Reviewer #2

(Remarks to the Author)

Reviewer #3

(Remarks to the Author)

The authors have satisfactorily addressed the reviewer's concerns. Therefore, the reviewer would recommend the publication of the paper.

Reviewer #4

(Remarks to the Author)

The authors have significantly strengthened their manuscript by including additional experiments and toning down the language. I would encourage the authors to do a last review for typos (e.g., "resulting in significant environmental contamination and leveraging carbon emission" in the Abstract should probably be "resulting in significant environmental contamination and greenhouse gas emissions."). Otherwise, this paper is now suitable for publication.

Responses to Reviewers' Comments

Reviewer #1 (Remarks to the Author):

One of the first and most obvious problems is the title: it has a key word "reprocessable" which readers will take as the most dominant feature of the reported bioplastics. However, throughout the entire writings, the word "reprocessable" occurs ONLY once, in the figure 1 legend. In other words, the main text of the whole paper never mentions the word "reprocessable" -- then why it is in the title?!

Response: We have replaced “Reprocessable” with “Recyclable”. The revised title is: “Sustainable DNA-Polysaccharide Hydrogels as Recyclable Bioplastics”.

The manuscript presents an interesting approach to developing sustainable bioplastics by leveraging the reversible imine bonding between oxidized polysaccharides (dextran, alginic acid, and carboxymethyl cellulose) and biomass-derived DNA. The authors demonstrate multi-closed-loop recyclability, water-based processability, biodegradability, and scalability, along with a life cycle assessment (LCA). The experimental data, including mechanical properties, biocompatibility, and environmental impact analysis, are in great amounts, well-documented, and most solid.

However, several aspects require correction, clarification, additional experimentation, and refinement to strengthen the claims and enhance the manuscript's scholarly impact. The innovative aspect also needs to be elaborate more. Below, specific concerns and comments are provided; they need to be addressed before the work can be published.

Response: We appreciate the reviewer's positive comments. We have addressed all the comments in the following point-by-point responses.

Major Concerns and Comments

1. Introduction, Rigor, and Novelty

- *Comment on Reversible Bonding and Polymer Breakdown: The authors write in the introduction: “formulas without reversible bonding features limit their potential in the breakdown of polymer chains, a critical factor for chemical recycling.” Polymer chain breakdown depends not only on reversible bonds but also on inherent degradability. This claim overstates the importance of reversible bonds and oversimplifies the complexity of polymer*

degradation. The authors should acknowledge that while reversible bonds (e.g., imine) facilitate chemical recycling, other factors like chain composition and environmental conditions (such as enzymes for bioplastics) also play significant, and might even be more important, roles. In addition, clarify how imine bonds specifically enhance recyclability compared to non-reversible bonds (both pros and cons).

Response: Indeed, we fully agree that polymer degradation depends on a multitude of factors, which include chain composition, polymer crystallinity, particle size and environmental factors, such as enzymes, exposure to sunlight and mechanical abrasion (e.g., through wave action). The cited publications present highly innovative perspective that inspired our current study. We would like to emphasize that our study introduces a complementary approach that differs from the previously reported methods, namely the crosslinking of polysaccharides and DNA through imine bond formation. The reversibility of imine bonds, and the possibility of hydrolysis to form aldehyde and amine groups in the presence of water, allows the resulting polymer to be (i) degraded and the resulting constituents recovered for recycling into products of alternative forms (Figures 2d) or (ii) reprocessed through aqua-healing (Figure 3b). Also, the use of DNA to crosslink polysaccharides, in our systems, allows the resulting polymer to be highly biodegradable, producing non-hazardous, environmentally benign products (Figure 3h). In contrast, using crosslinkers containing non-reversible bonds may form persistent polymer fragments even if the polysaccharides can be naturally degraded, which can take the form of micro/nano-plastics, requiring prolonged durations to naturally degrade in the environment.

We have amended the text in the introduction section in response to this important comment and further highlighted the unique advantages of our system: both components are naturally occurring biopolymers, unlike PEG-diacrylate (ref 24) and the epichlorohydrin-derived crosslinked DNA materials (ref 25), both of which require petroleum-based components.

Revisions have been introduced at the end of the 1st paragraph, Page 3 and at the end of 3rd paragraph, Page 5.

• *Eco-Friendliness Claim: In the authors' critique of prior DNA-based bioplastics (e.g., ref 24) as "not eco-friendly" ("...the chemical synthesis process is not eco-friendly, involving mass usage of organic solvents and relatively harsh synthesis conditions, e.g. a relatively high temperature under an inert gas environment"), the word of "chemical synthesis process" is ambiguous at least; if the ref24 is the target, then the statement is wrong. Please note that the*

authors' usage of sodium periodate (a hazardous chemical) in their process isn't entirely free of environmental concerns either. Thus, a blanket of assessment of "not eco-friendly" is not appropriate here. It is suggested that the authors should soften this critique, and explicitly and fairly compare their method with that of ref 24 (e.g., solvent use, temperature, energy input, etc.).

Response: As suggested, we have softened this critique and made a fair comparison supported with data details. For example, in ref 24, while it is highly innovative, the synthesis process could be more sustainable, as it involves the extensive use of organic solvents and relatively harsh synthesis conditions and/or relies on key components derived from unsustainable petroleum feedstocks. These increase their potential carbon footprint and environmental impact. This has been introduced into the end of 1st paragraph, Page 3.

In addition, previous work (ref 24) to crosslink DNA and ionomers involves the mass use of N,N-dimethylformamide (a 10-to-1 mass ratio to ionomers), a relatively high reaction temperature of 130 °C, and an inert gas environment of nitrogen. In comparison, the reaction in the present work is water-based and proceeds at room temperature under ambient atmosphere, thereby reducing the energy consumption and potential negative environmental impacts during the synthesis process. This issue has been addressed by adding an appropriate statement in the middle of 1st paragraph, Page 4.

Indeed, the need for a cost-effective "clean" source for NaIO₄ is a limiting factor in the current process. The use of solar light to produce IO₄⁻ or, alternatively, the solar-light photocatalyzed oxidation of the polysaccharides to the aldehyde-modified polysaccharides could potentially overcome this limitation. Photocatalyzed oxidation of iodide to periodate in aqueous basic solutions, or photocatalyzed oxygen-mediated Fenton oxidation of I⁻, could provide parallel close-loops for the present process, synthesizing the bioplastic materials. This has been discussed in the Conclusion section, Page 14.

• *Similarity wording ("unintentional plagiarism"?) to Ref 24: The sentence about biomass DNA availability mirrors ref 24 too closely without acknowledgement. It requires either a citation of ref24, or rephrasing to avoid unintentional overlap.*

...DNA on the earth is estimated at 50 billion metric tons, and less than 1% of this biomass DNA reserves is more than enough to meet the annual demand on the feedstock of worldwide commodity plastics production

...DNA on earth is estimated to be 50 billion metric tonnes; less than 1% of it would be more than sufficient to fulfil the annual demand on the feedstock of global commodity plastic production.

Response: We have rephrased this sentence to eliminate unintentional overlap and added a citation of ref 23,24 (the middle of 1st paragraph, Page 3).

• *Novelty* The author's work is clearly a nice extension/addition of a previous reported work (ref24) with the incorporation of polysaccharides into bioplastics using a different, imine-bond crosslinking method. Please note that the polysaccharide-based bioplastics have been extensively reported for many years, thus, the novelty of this work over ref 24 (PEGDA-crosslinked DNA plastics) and other polysaccharides-based plastics needs sharper delineation and more extensive elaboration — e.g., emphasize the water-based process, more contents than DNA alone, better properties than polysaccharides, and potential synergy between DNA and polysaccharides --- to showcase DNA-polysaccharides' distinct advantages.

Response: As suggested, we have made corresponding revisions to polish and further elaborate the novelty of this work by comparing it with the ref. 24 PEGDA-DNA and other polysaccharides-based plastics, such as water-based process, mild reaction under ambient conditions. Revisions have been made to the 1st paragraph of Page 4.

In addition, incorporating DNA into the bioplastic matrix offers a key advantage by introducing an additional, orthogonal degradation pathway beyond those available to polysaccharide-based systems alone. While enzymatic degradation of polysaccharides—such as dextran hydrolysis by dextranase—is well-established, these pathways are typically limited to specific glycosidic linkages. DNA, in contrast, is easily degraded by a distinct set of enzymes (e.g., DNase), which are abundant in many natural environments. This approach not only broadens the environmental triggers for material degradation, but also, enables more tunable and potentially faster degradation pathways. Moreover, the inclusion of DNA adds chemical diversity to the bioplastic, offering additional handles for molecular design and programmability that are not easily accessible with polysaccharides alone. Discussion has been added to the 2nd paragraph of Page 10.

2. Chemistry and Characterization

• *Oxidation Reaction Misnomer:* The periodate oxidation specifically refers to Malaprade reaction (L. Malaprade. Action of polyalcohols on periodic acid. Bull. Soc. Chim. France. 1928,

43: 683.; L. Malaprade. Oxidation of some polyalcohols by periodic acid-applications. *Compt. Rend.* 1928, 186: 382.). Criegee oxidation usually refers to the oxidation of vicinal diol using lead tetraacetate as the oxidant. The authors should correct this throughout the manuscript (e.g., Fig. 1, Methods, etc.) to ensure accuracy. The Malaprade reaction cleaves a carbon-carbon bond of vicinal diol structure, dextran or other polysaccharides are not exceptional. In Fig. 1, it looks like that the authors presented rearrangement forms of oxidized dextran, that should be improved and/or clarified as well.

Response: As suggested, the Criegee oxidation has been corrected to the Malaprade reaction. Also, Fig. 1 has been revised accordingly (Page 4).

• *Imine Bond Evidence:* The NMR evidence (Fig. S1) for imine bond formation is promising but insufficiently conclusive. Why not IR spectroscopy? IR could confirm the C=N stretch and provide evidence about hydrogen bonding interaction between DNA and polysaccharides. Additionally, the amine group from A, T, G, C of DNA has different nucleophilic activity. Usually, the amine group from G has the highest reactivity, but T definitely cannot react with aldehyde to form imine bond because the nitrogen atom in organic compounds can form a maximum of three valence bond according to valence bond theory. This should be corrected in Fig. 1. The authors speculate multiple reactive sites (e.g., N2 of guanine, N6 of adenine) but provide no confirmation. It needs to confirm/clarify which base reacted with the aldehyde on oxidized polysaccharides.

Response: As suggested, we conducted the FTIR measurement and added the result in the Revised Manuscript and Supplementary Information. In summary, the IR spectra of dextran (Dex), oxidized dextran (ox-Dex), DNA, Dex-DNA (original), and Dex-DNA (recycled) have been shown in the newly added Fig. S11. After oxidation, the ox-Dex exhibits a low intensity signal at 1732 cm^{-1} (corresponding to the -C=O stretching from an aldehyde group), comparing to that of Dex (*Polymer* **2005**, 46, 9604). In the IR spectrum of DNA, the -C=O stretching vibration (1650 cm^{-1} , 1631 cm^{-1}) and -N-H stretching vibration (3332 cm^{-1} , 3210 cm^{-1}) are observed. Upon the formation of Dex-DNA composite, -C=N stretching vibration of Schiff base exhibits a characteristic peak at 1654 cm^{-1} . This peak is also observed in the recycled Dex-DNA bioplastic. Compared to the phosphate (PO_2^-) stretching vibrations in pure DNA, the 1215 cm^{-1} peak intensities of Dex-DNA decrease after Schiff base formation due to electronic redistribution and molecular interactions. This discussion has been added as the 3rd paragraph of page 5.

In addition, in the ^1H NMR spectrum of oxidized Dex (Fig. S1), the aldehyde group shows a characteristic signal at $\sim\delta$ 9.7 ppm (*Polymer* **46**, 9604-9614 (2005)). No characteristic signal is observed at this region from natural-derived DNA (Fig. S2). Upon the formation of Dex-DNA, an additional signal is obtained at δ 9.2–9.3 ppm (Fig. S3), suggesting the formation of imine groups ($\text{RN}=\text{CHR}'$), which is also observed in recycled sample (Fig. S4). As amine group is on N6, N4 and N2 positions of adenine (A), cytosine (C) and guanine (G), respectively, but not thymine (T) (Fig. 1a i), to determine which bases interact with the aldehyde groups of Dex, oligomers of nine bases (A9, C9 and G9) were used, respectively, to react with Dex. Compared to the ^1H NMR spectra of A9, C9, G9 (Fig. S5, S6, S7) and Dex (Fig. S1), the formed Dex-A9, Dex-C9 and Dex-G9 exhibit additional signals at $\sim\delta$ 9.25, 9.26, 9.27 ppm (Fig. S8, S9, S10), respectively, corresponding to the formation of imine groups. These results have been added to the 2nd paragraph of page 5.

3. Water Processability and Resilience

• *Solvent-Free Claim: The authors claim “solvent-free water (re)processability,” but water is a solvent; thus, the claim of solvent-free is misleading. It should be deleted.*

Response: As suggested, we have deleted “solvent-free” in the Revised Manuscript.

• *Water Resilience Concern: Water processability is a double-edged sword: it implies poor moisture resistance, a known challenge for bioplastics. The authors tout this as an innovation/advantage, but applications requiring water stability (e.g., packaging) may be limited. This point should be discussed (pros and cons), and overselling should be avoided.*

Response: We agree that both pros and cons come together with water processability. It is worth mentioning that water processability, including water-based molding and aqua-healing, is a relatively eco-friendly approach. However, this also indicates the relatively poor moisture resistance of the produced bioplastics. Thus, for practical applications, such as packaging materials where high moisture resistance is necessary, further processing of the Dex-DNA bioplastics is, indeed, required. Potential methods include applying moisture-resistant layers, such as hydrophobic coating (e.g., polydimethylsiloxane), to these bioplastics. This discussion has been added at the end of 2nd paragraph, Page 8.

4. Mechanical Properties and Recycling

• *Imine Bond Stability Post-Recycling: The mechanical properties remain stable after 10 cycles (Fig. 2f), which is impressive. To directly link the reversible chemistry to recyclability, NMR/IR*

should be used to track imine bond integrity post-recycling. This could strengthen the multi-closed-loop claim.

Response: We conducted both ^1H NMR and FTIR measurements and added the results in the Revised Manuscript and Supplementary Information. In the ^1H NMR, the imine signals at δ 9.2–9.3 ppm are observed in both original and recycled Dex-DNA bioplastics (Fig. S3, S4). The IR spectra, including both the original and the recycled samples, were also recorded and displayed in the newly added Fig. S11. The $-\text{C}=\text{N}$ stretching vibration of Schiff base exhibits a characteristic peak at 1654 cm^{-1} , which is observed in both samples. These results demonstrate the recyclability via imine bonds, strengthening the multi-closed-loop claim. The discussion has been added in the 2nd paragraph, Page 6.

• *Solvent Resistance: The bioplastics resist hexane, chloroform, and DMSO (Fig. 3a) but dissolve in extreme pH (Fig. S6). What about the solubility in glycerol/ethylene glycol (Polar but viscous; may act as plasticizers), and formamide (highly polar)? which are other common relevant organic solvents. These testing are optional but if conducted, hopefully will broaden the resistance profile, adding more evidence to strengthen the resistance claim.*

Response: As suggested, we conducted the solvent resistance test in glycerol, ethylene glycol, and formamide. We found that after 7 days, the 10%Dex-5%DNA bioplastics were fully dissolved in ethylene glycol, while being significantly softened/swollen in glycerol or formamide. We have added these results in the revised Supplementary Information (Fig. S20 and Movie S1-2) and discussed the results in the Revised Manuscript (2nd paragraph, Page 8).

5. Life Cycle Assessment (LCA)

The authors present a cradle-to-grave LCA to evaluate the environmental impacts of DNA-polysaccharide bioplastics, which is commendable for its effort and aligning with sustainability goals, but it is quite premature for conclusive or even informative results because the reported bioplastics is still at a very early stage of the lab-scale: there is no large scale production, the entire process is not optimized, and the industry-scale parameters (e.g., energy efficiency, materials yields, specific recycling infrastructures, etc.) are all speculative. The authors should trim down significantly the entire LCA sections and emphasize the preliminary and speculative nature of this exercise and the non-conclusive characteristics of results (e.g., is DNA production really contributing to 99% Ozone depletion? Is this conclusion reliable/convincible?)

Response: We agree with the reviewer that the LCA for this emerging technology at the early stage may not be subject to iterative optimization yet. However, this is also the valuable part of LCA that can provide the preliminary environmental impacts of emerging products for further improvement. As stated in the recent article by Chen et al. (*Nat. Common.* 2025, 16, 1), early-stage LCA can effectively bridge the molecular and process levels to rigorously quantify the impact of different systems.

For the ozone depletion contribution, the primary contributor is the chloroform (4L) used in 1 kg DNA production (*Nat. Protoc.* 2006, 1, 2320; *J. Cotton. Sci.* 2000, 4, 193; *J. Cytol.* 2019, 36, 116). Based on the data given by Ecoinvent 3.9 database, the ozone depletion potential of 1 kg chloroform is 7.3×10^{-4} kg CFC-11eq, which is ~2000 times higher than other substrates used in the system. This significant contribution is due to the tetrachloromethane emission during the chloroform production. This has also been widely reported by previous studies (*Clean. Prod.* 2023, 414, 137573; *Nat. Geosci.* 2019, 12, 89).

Following the reviewer's comments, we have addressed these issues in the revised manuscript and trimmed down significantly the LCA sections (Page 13 and the 1st paragraph of Page 14).

More specifically:

1. *There exist huge data extrapolation risks when conducting cradle-to-grave LCA at this early stage.*
2. *Feedstocks are entirely unknown.*
3. *Recycling assumptions are also extremely speculative as there are no insights whatsoever at this stage how the recycling will be accomplished.*
4. *Lab scale recycling (shaking etc.) isn't the same as real-world recycling, which includes contamination, extreme and uncontrollable weather, UV, etc.*

Response: 1. We fully acknowledge that conducting a cradle-to-grave LCA at the early stages of the bioplastic development involves significant uncertainties and data extrapolation risks. The purpose of our *ex-ante* LCA is to provide a preliminary baseline assessment of the environmental performance of the proposed bioplastic compared to conventional plastics. This early-stage analysis serves as a directional guide for process optimization and design improvement, rather than a definitive environmental impact statement.

2. The feedstock for DNA extraction is assumed to be waste plant leaves, which reflects a plausible, low-impact biomass source for early assessment purposes. We have acknowledged that this may evolve with future supply chain development.

3. We agree that recycling assumptions are highly speculative at this stage. We therefore refrained from including detailed recycling scenarios and instead noted that the recyclability of the bioplastic would depend on future technological developments and infrastructure compatibility.

4. We acknowledge that lab-scale recycling (e.g., via solution processing or mechanical agitation) does not fully reflect the complexities of real-world conditions, including contamination, UV exposure, and environmental stressors. These factors are flagged in the manuscript as limitations that future LCA iterations should incorporate as data become available.

Lastly, the inventory data of the cradle-to-grave *ex-ante* LCA were majorly upscaled based on the experimental data that can be further improved or optimized along the commercialization as the investigated technologies were at the early stage.⁴⁰ We have added clarifying text in the Methods (2nd paragraph, Page 18) and Discussion section (1st paragraph, Page 14) to reflect these limitations and forward-looking considerations.

Recommendation

This is clearly a nice piece of research that pushes DNA-based materials broader and further. Data is solid and supports most of the conclusions (except the LCA work). Major revisions, mostly on writing, are needed; especially, some overstated claims need tempering.

Response: We sincerely appreciate the detailed, highly professional, and insightful comments of the reviewer, all of which have been addressed in our responses above. We certainly appreciate the efforts of the reviewer to improve our paper.

Reviewer #2 (Remarks to the Author):

Response: We sincerely appreciate your positive comments. We have responded to all your suggestions in a point-by-point manner, as detailed in the “Response to Reviewer 1”.

Reviewer #3 (Remarks to the Author):

The development of sustainable plastics is crucial for the sustainable development of human society. Although great efforts have been made and various sustainable plastics have been reported, the development of DNA-based bioplastics is still an exciting advance. In this paper, the authors demonstrate a facile way to produce bioplastics derived from biomass DNA and polysaccharides. These bioplastics exhibit attractive properties, such as water processability, biodegradability, good recyclability, scalability, high manufacturing precision, biocompatibility, etc., and also can minimize the risk of microplastic pollution and reduce carbon emissions. These results are impressive, and certainly of interest for the readers of Nature Communications. Therefore, I would like to recommend the publication of this manuscript after addressing the following concerns.

Response: We sincerely appreciate the positive comments stated by the reviewer.

1. When competing with conventional plastics, the cost of the material is also a very important issue. It would be better to add a discussion of the potential cost of the biomass-based bioplastics in large-scale applications.

Response: As suggested, we conducted the calculation and discussed the estimated cost of materials. We found that based on the method in our study, the cost of raw bioplastic materials ranges from approximately USD 3.1 to 5.2 per gram. To move toward future large-scale production, it is, however, essential to consider cost-reduction strategies, including more affordable extraction techniques, optimization of raw material supply chains, and the use of more energy-efficient fabrication equipment than those employed in our current experiments. In response, we have added the calculation details in Table S2 (Supplementary Information) and added a corresponding discussion in the Revised Manuscript (end of the 1st paragraph, Page 14).

2. As the carrier of genetic information, DNA derived from different sources may have different sequences and structures. Would this affect the properties of the resulting bioplastics? And, is it possible to utilize the genetic information in the production and application of the DNA-based bioplastics?

Response: Indeed, DNA extracted from different biological sources may differ in sequence, length, and secondary structure. These variations could affect the resulting bioplastic due to difference in imine bond densities, and consequently, influence their mechanical properties (e.g., Young's modulus). However, key characteristics such as enzymatic degradability and

closed-loop recyclability are expected to remain similar, as they are governed by the polymeric backbone of DNA rather than its specific sequence. Nevertheless, harnessing pure sequence-programmed DNA stocks could be produced by genetically-engineered crops. While such approaches are at present cost-prohibitive and impractical for large-scale sustainable material production, for special material applications, adaptation of these concepts might be interesting. The issues raised by the reviewer were addressed by adding a short paragraph to the conclusion section.

3. The fact that the structure of the bioplastics can be affected by water also makes the application of these bioplastics very limited. Is it possible to find a way to solve this problem?

Response: It is worth mentioning that water processability, including water-based molding and aqua-healing, is a relatively eco-friendly approach. However, this also indicates the relatively poor moisture resistance of the produced bioplastics. Thus, in practical applications such as packaging materials where high moisture resistance is necessary, further processing of the Dex-DNA bioplastics is required. Potential methods include applying moisture-resistant layers, such as hydrophobic coating (e.g., polydimethylsiloxane), to these bioplastics. This discussion has been introduced to the end of 2nd paragraph, Page 8.

4. In Fig. 3f, the authors demonstrated the production of a medical microneedle patches with the bioplastics, which is very interesting. Could the microneedles penetrate the stratum corneum?

Response: To address this point, we conducted an experiment to evaluate whether the bioplastic microneedles could penetrate the stratum corneum. The results confirmed successful penetration. We have included the corresponding data as Fig. S22 in the Supplementary Information and added a brief discussion in the 1st paragraph of Page 10 in the Manuscript.

5. The authors demonstrated a sequential molding and drying process to form molded bioplastic products from hydrogels. A more detailed discussion of the possible molding mechanism is suggested.

Response: To provide a clearer and more detailed discussion of the molding mechanism, we have added a schematic illustration of the process (Fig. S15, Supplementary Information, mentioned, also, in text). As described, during room-temperature drying, the gradual evaporation of water enables polymer chains within the hydrogel to reorganize and deposit along the mold boundaries. This confined drying promotes the formation of a dense, shaped

bioplastic structure, as the capillary forces and polymer interactions guide the hydrogel shrinkage and consolidation within the mold cavity. A brief discussion has been added in the last paragraph, Page 6.

6. There are some grammatical errors and typos in the manuscript, for example on page 2 line 58-59, “and and contributed significantly to carbon emissions”.

Response: We have corrected the identified typo and thoroughly reviewed the manuscript to address other grammatical errors, typographical mistakes, and formatting issues.

Reviewer #4 (Remarks to the Author):

The article “Sustainable DNA-Polysaccharide Hydrogels as Reprocessable Bioplastics” describes the synthesis of a biodegradable hydrogel from dextra, alginic acid, carboxymethyl cellulose, and DNA. While the material seems innovative, the authors make a number of broad statements with little experimental evidence (see below). I would suggest major revisions before the article can be considered suitable for publication.

1. Paragraph 2 of the Results and discussion section reads more like an introduction or conclusion paragraph and should be removed given that no experiments are discussed to support the claims.

Response: Following the comment of the reviewer, the “Results and Discussion” section was re-edited to be focused on the experimental concept and experimental results, Page 4.

2. More material characterization should be provided, both of the original and recycled polymer (e.g., barrier properties, melting point). How do these properties link to the suggested applications of biomimetic surfaces, micro-needles, and photonic nano-patterns? Furthermore, how do the measured tensile properties compare to other bioplastics or conventional plastics?

Response: As suggested, we have significantly expanded the material characterization of both the original and recycled bioplastics. The additional experimental results have been incorporated into the Revised Manuscript and Supplementary Information (SI).

Specifically, we now include:

- Chemical structure characterization via ¹H NMR (Fig. S1–S10, SI) and FTIR (Fig. S11, SI). (Appropriate revisions were introduced to the 2nd and 3rd paragraph of Page 5);
- Barrier performance, including water vapor transmission rate (WVTR) and permeability (*P*) tests (Fig. S16, SI) (1st paragraph of Page 7) were added;
- Thermal properties, such as melting point determined by DSC (Fig. S17, SI) (1st paragraph of Page 7);
- Mechanical testing, including tensile properties comparison (Fig. S19, SI) (1st paragraph of Page 8) and microneedle penetration into mouse stratum corneum (Fig. S22, SI) (1st paragraph of Page 10);
- Chemical resistance to organic solvents (Fig. S20; Movies S1–S2, SI) (2nd paragraph of Page 8).

The additional data confirm that the recycled bioplastics retain key structures and functional properties, including similar WVTR and melting points to the original material. The

bioplastics-based microneedles were shown to successfully penetrate the mouse stratum corneum, which might be further developed as monitoring and delivery devices.

Additionally, we have added an analysis regarding the tensile property (Fig. S19, SI) by comparing it with related research of DNA-based and water-processable plastics, as well as several typical commercial plastics. We found that the bioplastics exhibit a Young's modulus range that overlaps with those of certain reported and commercial plastics, such as HDPE and LDPE. A discussion addressing these issues has been introduced to the 1st paragraph of Page 8.

3. Is there an ASTM standard that could be used for measuring biodegradability, rather than the relatively uncontrolled experiment presented in Fig. 3?

Response: In this study, we employed two complementary approaches to evaluate biodegradability:

- a. Natural degradation under Singapore's environmental conditions (Fig. 3h)**, with detailed documentation of local weather parameters (e.g., temperature of 23-34 °C, humidity of 80-90%). This method is also applied in prior literature (e.g., *Nat. Sustain.* 2021, 4, 627; *ACS Nano* 2022, 16, 16414; *Adv. Mater.* 2023, 35, 2301398) and provides practical insights into material behavior in real-world settings.
- b. Controlled enzymatic degradation assays in the lab (Fig. 3i and S23)**, which offers reproducible and quantitative data. This approach is also widely adopted in related studies (e.g., *J. Am. Chem. Soc.* 2020, 142, 10114; *J. Am. Chem. Soc.* 2021, 143, 19486) and serves as a rigorous supplement to environmental testing.

Nevertheless, we fully acknowledge the importance of standardized testing methods. Accordingly, we have commissioned an independent third-party to test our bioplastics using relevant ASTM standards. The results will be included in the Supplementary Information as soon as they become available (in about 6 weeks).

4. Language should be toned down. For example, a 500 mL batch is certainly up-scaling the process, but should not be considered "mass-producing".

Response: We have replaced "mass production" with "upscaling production" throughout the manuscript. In addition, we have moderated the language concerning the eco-friendliness and

moisture-resilience features, as detailed in our responses to Reviewer 1 under the comments “Eco-Friendliness Claim” and “Water Resilience Concern.”

5. *The presentation of the LCA results in the abstract, introduction, and conclusion are misleading. Upon reading the results section, it seems that the 59-78% reduction in GHG emissions relative to conventional plastics is only achieved with the bioplastic is recycled 10 times. Please include this important information in the abstract, introduction, and conclusion. Also consider running a scenario in which conventional plastics such as PET or HDPE are mechanically recycled 10x.*

Response: As suggested, we have included the important information (recycled 10 times) throughout the manuscript. We have also run a scenario where the conventional plastics are mechanically recycled 10x and compared with the current results (Fig. S29, Supplementary Information). After 10 recycling cycles, our bioplastic exhibits a comparable global warming potential (GWP) to that of commercial plastics. As the current production process stays at the laboratory scale, we anticipate that further reductions in life-cycle greenhouse gas emissions can be achieved through process optimization and the use of sustainably sourced raw materials at scale. This discussion has been introduced in the 1st paragraph, Page 14.

6. *What is the justification for using a volumetric functional unit for the LCA (m³) rather than the more common mass-based functional unit (kg or metric ton)?*

Response: Our bioplastics have a much lower density (~150 kg m⁻³) compared to the currently conventional plastics. Given the same functional shape, the usage of bioplastics will be lower than the conventional plastics. Hence, it will be fair to compare the environmental impacts based on the volume basis rather than the weight basis. To clarify this point, we have revised the contents accordingly by providing the justification in Method section (2nd paragraph, Page 18).

7. *Please check for typos throughout. For example, on page 4, “to form sold bioplastic products” should be “to form solid bioplastic products”.*

Response: We have thoroughly checked the manuscript and typos have been corrected.

Responses to Reviewers' Comments

Reviewer #1 (Remarks to the Author):

The authors have responded to my criticisms professionally and thoughtfully, and I am excited to recommend it for publication.

That being said, I continue to have a concern about the LCA, which should not affect my recommendation; rather, it serves as a documented record of my concern and also as a suggestion to further enhance the paper's rigor and impact.

The authors' argument that their LCA can provide a "preliminary baseline" for process optimization is noted, and I understand that the authors are trying to connect their new material closely and quantitatively with sustainability through LCA. However, the problems are:

1) The value of any LCA hinges on the reliability and process-relevance of the underlying data. At this stage, the authors' data lack the maturity to support meaningful environmental impact projections. Their statement that the technology is "...not subject to iterative optimization yet" further underscores this issue. Such ex-ante LCA will be very limited in its utility as a future "guide"; rather, premature LCA is more likely to risk producing false precision and misleading guidance.

2) While LCA can theoretically be applied to any new and old material/process, ex-ante LCA is most valuable as a dynamic tool within active process development where assumptions can be iteratively refined. However, once published in peer-reviewed literature, these preliminary, or worse, premature analyses become static references that may be cited without full appreciation of their underlying limitations, particularly as the foundational assumptions, including processing and recycling methods, become established and/or outdated.

3) The paper itself without LCA is good enough to be published – the material's synthesis, characterization, degradation, and applications are all quite impactful. However, adding this ex-ante LCA raises questions, reduces readers' focus, and diminishes the paper's rigor.

Suggestions (my personal preference, no need to follow):

remove LCA from the abstract. In the discussion section, discuss the sustainability/recycling aspects of the material and only include some of the LCA data in the supplementary materials."

Response: We highly appreciate the professional comments and recommendation from the reviewer. We agree with the reviewer on the preliminary perspective and role of the *ex-ante* LCA in new material development and the iterative process to optimize and improve the new material production. Following the suggestion from the reviewer, we have revised the manuscript accordingly. Specifically, the LCA in the abstract has been removed. In the discussion section, we have shortened the discussion part related to LCA results and moved the original Fig. 5f to supplementary information.

Reviewer #2 (Remarks to the Author):

Response: We appreciate the reviewer's support and suggestive comments.

Reviewer #3 (Remarks to the Author):

The authors have satisfactorily addressed the reviewer's concerns. Therefore, the reviewer would recommend the publication of the paper.

Response: We appreciate the reviewer's support.

Reviewer #4 (Remarks to the Author):

The authors have significantly strengthened their manuscript by including additional experiments and toning down the language. I would encourage the authors to do a last review for typos (e.g., "resulting in significant environmental contamination and leveraging carbon emission" in the Abstract should probably be "resulting in significant environmental contamination and greenhouse gas emissions."). Otherwise, this paper is now suitable for publication.

Response: We appreciate the reviewer's support and suggestive comments. Indeed, "greenhouse gas emissions" is more appropriate. We have thoroughly checked the manuscript and corrected the typos.